# Hybrid Instruments Network Optimization for Air Quality Monitoring

Nishant Ajnoti[1], Hemant Gehlot[1], Sachchida Nand Tripathi[1,2]

[1]Department of Civil Engineering, Indian Institute of Technology Kanpur, India
[2]Department of Sustainable Energy Engineering, Indian Institute of Technology Kanpur, India
*Correspondence to*: Hemant Gehlot (hemantg@iitk.ac.in)

**Abstract** The significance of air quality monitoring for analyzing the impact on public health is growing worldwide. A crucial part of smart city development includes deployment of suitable air pollution sensors at critical locations. Note that there are various air quality measurement instruments ranging from expensive reference stations that provide accurate data to low-cost sensors that provide less accurate air quality measurements. In this research, we use a combination of sensors and monitors, which we call hybrid instruments and focus on optimal placement of such instruments across a region. The objective of the problem is to maximize a satisfaction function that quantifies the weighted closeness of different regions to the places where such hybrid instruments are placed (here weights for different regions are quantified in terms of the relative population density and relative $PM_{2.5}$ concentration). Note that there can be several constraints such as those on budget, minimum number of reference stations to be placed, set of important regions where at least one sensor should be placed and so on. We develop two algorithms to solve this problem. The first one is a genetic algorithm that is a metaheuristic and works on the principles of evolution. The second one is a greedy algorithm that selects the locally best choice in each iteration. We test these algorithms on different regions from India with varying sizes and other characteristics such as population distribution, $PM_{2.5}$ emissions, budget available, etc. The insights obtained from this paper can be used to quantitatively place reference stations and sensors in large cities rather than using ad hoc procedures or rules of thumb.

## 1 Introduction

According to the World Health Organization (WHO), ambient air pollution is a significant threat to people's health, causing around 6.7 million premature deaths annually in 2019 (Fuller et al., 2022). Shockingly, 99% of the global population resides in areas that don't meet WHO's air quality guidelines, with 89% of these premature fatalities occurring in low or middle-income countries (WHO, 2022; Pandey et al., 2021). To address this issue, it's crucial to develop suitable sensor networks by putting the air pollution monitors or sensors at appropriate locations, meeting the requirements of various groups in the city, and providing much-needed information. Air pollutant concentrations have traditionally been monitored using reference stations (we will refer to them as monitors in this paper) which are highly accurate but also very costly, limiting their widespread deployment (Lagerspetz et al., 2019). To achieve accurate air pollution monitoring within metropolitan regions, hundreds or

even thousands of reference stations are required, which proves costly to maintain and operate (Zikova et al., 2017). However, the emergence of low-cost air quality sensors presents an opportunity for higher-density deployments and improved spatial resolution in monitoring (Spinelle et al., 2017; Castell et al., 2017). Low-cost sensors offer a cost-effective solution, reducing installation and maintenance expenses and facilitating broader spatial coverage, particularly in remote areas. Therefore, in order to balance the accuracy of monitoring along with costs involved in such instruments, we will consider deployment of both monitors and sensors in this paper.

Some studies focus on optimizing air quality monitoring networks (AQMNs) using different models: physical models (Araki et al., 2015; Hao and Xie, 2018) and learning-based models (Hsieh et al., 2015). However, the accuracy of these methods relies heavily on the precision of the air quality models, and both Hao and Xie (2018) and Hsieh et al. (2015) required existing air quality measurements as inputs for their prediction models which largely depend on the quality and completeness of input data. The studies by Li et al. (2017), Brenzia et al. (2015), and Zikova et al. (2017) discuss ad-hoc placement of air quality sensors in their respective study regions or using some rules of thumb. But this shows that the placement of sensors is not optimized under the budget constraints that might be present. To address these challenges, it becomes crucial to develop more strategic approaches for placing air quality sensors. Properly optimized sensor placement can lead to a more comprehensive and accurate understanding of air pollution patterns, facilitating targeted pollution control measures and ultimately improving public health and environmental management.

Lerner et al. (2019) present a method for optimizing sensor placement based on sensor characteristics and land use analysis. Sun et al. (2019) also propose an optimal sensor placement strategy based on population density without relying on air pollution data. Their study highlights that humans naturally depend on the closest station to observe and obtain relevant information regarding the environment when multiple stations are present in a city. The satisfaction regarding the information increases as one moves closer to the adjacent station. Unlike Lerner et al. (2019), Sun et al. (2019) represent the benefit of placing a sensor in a particular grid to the citizens not just living in that grid but also to those living the nearby grids. However, Sun et al. (2019) has limitations in that it does not incorporate air pollution data as a parameter in optimization, which raises concerns about the accuracy and reliability of the obtained results. Furthermore, both Lerner et al. (2019) and Sun et al. (2019) only consider deployment of one type of sensor but as we discussed previously, both monitors (that are very accurate) and sensors (that are not that accurate but much more economical than monitors) should together be considered for deployment.

Note that Castell et al. (2017) also highlighted that sensors alone may not provide accurate air quality measurements as compared to reference instruments or monitors. Our proposed approach aims to leverage the strengths of both sensors and monitors to enhance air quality monitoring in a cost-effective manner. We propose to develop a framework for placing hybrid instruments with the objective of maximizing the public satisfaction by considering emission spread and population density as parameters (while considering the benefit of placing instruments in nearby grids and not just the grids where they are placed).

Also, several notable constraints such as having at least one sensor in a given set of important grids (like important residential or commercial areas), not having monitors in certain given grids (like places with sparse population, water bodies, etc.), having a minimum number of grids where monitors should be placed in the network, etc., have been proposed in the optimization formulation. Therefore, the following are the contributions of our work:

- Our research focuses on optimal deployment of hybrid air-quality monitoring networks consisting of monitors and
sensors where the goal is to maximize public satisfaction by providing accurate air quality information while considering several budget and other constraints.

- We propose a Genetic algorithm (GA) and a greedy algorithm (GrA) to solve the developed optimization problem.

- We test the developed algorithms on networks of varying sizes and geographic locations.

This paper's remaining sections are organized as follows: Section 2 describes the optimization problem and presents the
algorithms for solving the problem. The next section provides the numerical results tested using different algorithms under different settings. The final section concludes our study and provides future directions.

## 2 Methodology

This section is divided into two parts. The first part describes the problem statement for optimization of a hybrid instrument network. The second part describes the methods proposed to solve the optimization problem. The second part is further sub-
divided into two sub parts: GA and GrA respectively.

### 2.1 Problem Statement

Our approach focuses on placing sensors and monitors in order to maximize a utility function quantifying popular satisfaction with the instrument placements. Realising that humans naturally depend on the closest station to observe and obtain relevant information regarding the environment when multiple stations are present in a city, we assume that an individual's satisfaction
$g(d)$ is a function of his or her distance $d$ to the closest sensor or monitor (Sun et al., 2019). Intuitively, the satisfaction with the information increases as one moves closer to the adjacent station. That is because people will have higher confidence on the readings by sensors or monitors that are closer to them rather than readings from instruments that are farther from them. Therefore, $g(d)$ must satisfy the following conditions as stated in Sun et al. (2019): (i) $g(d)$ must be a decreasing function, i.e., for any $d1 \leq d2, g(d1) \geq g(d2)$, (ii) for any $d \geq 0, g(d) \geq 0$ and $g(0) = 1$. The foremost condition corresponds
to the relation of satisfaction function with distance, while the latter ones assure the fact that the $g \in [0, 1]$ and $g$ is the highest

when the distance is zero. The following exponentially decreasing function $g(d)$ readily satisfies the aforementioned conditions (Sun et al., 2019):

$$g(d) = \exp\left(-\frac{d}{\theta}\right), \tag{1}$$

where $\theta$ is an exponential decay constant[1]. The exponential decay function is often chosen in similar studies and practical applications because of its simplicity and effectiveness in modelling the attenuation of signal or influence with increasing distance in studies such as Sun et al. (2019). It aligns with the intuitive idea that the influence of air quality monitoring decreases as one moves farther away from the monitor. We also present the results with another appropriate satisfaction function later. Note that monitors and sensors are not differentiated while determining the satisfaction function in our problem. That is because in many practical air quality monitoring scenarios, users may not be either interested or be able distinguish between the data collected from monitors and sensors (if the information related to the type of instrument is not openly available). From the user's perspective, the primary concern may be just to obtain reasonable air quality information, rather than worry about the specific source of the data.

In accordance with the standard procedure for environmental monitoring (Krause et al., 2008, Hsieh et al., 2015), we divide the city into distinct, equal-sized square grids. Then, we place our hybrid instruments (sensors and monitors) in these fragmented grids. Let $V = \{a | a = 1, 2 \ldots, n\}$ represent a set of grids in the interested geographical area, in which $n = |V|$ represents the total number of grids. For each $a \in \{1, 2 \ldots, n\}$, let $p_a$ represent the percentage of people living in grid $a$, $e_a$ represents the percentage of PM$_{2.5}$ emissions[2] in grid $a$ and $m_a$ denotes the weighted average of $p_a$ and $e_a$ of grid $a$, i.e., $m_a = (w_1 * p_a) + (w_2 * e_a)$, where $0 \leq w_1, w_2 \leq 1$ and $w_1 + w_2 = 1$. Note that both population density and PM$_{2.5}$ emissions are important factors while deciding the relative importance of various grids. Population density reflects the concentration of people residing in that grid, while the PM$_{2.5}$ emissions are an indicator of the level of fine particulate matter in the air within that grid (secondary aerosol production and pollution transport also play a role in the concentrations but they are not considered here due to lack of data). Doing a weighted average of the corresponding percentage values of these parameters provides a single value that quantifies the importance of a particular grid and allows comparing between different grids. Also, if we do not the weighted averaging, and individually minimize some metrics related to emission and population then it will result into a multi-objective optimization problem which is much more difficult to solve and analyze (Deb, 2001).

We will now introduce some variables to define the optimization formulation. The notations are summarized in Table A1 of Appendix A. Let $S$ be a set of grids where instruments (sensors and monitors) are placed (i.e., set $S$ consists of each grid $a$

---

[1] Depending on the largest distances that are considered in a grid network and the precision that is being considered, $\theta$ should be appropriately decided. For instance, if the computation precision being used is say about $10^{-5}$ and the largest distance is say 10 units then $\theta = 1$ might reasonable since $e^{-\frac{10}{1}} = 4.5 * 10^{-5}$.

[2] We acknowledge with the distinction between PM2.5 emissions and PM2.5 concentrations (which are to be measured by the network), with the possible impacts of secondary aerosol formation and pollution transport not being accounted for by using emissions information alone. In our approach, we initially prioritize PM2.5 emissions as the foundational data for instrument placement. However, the placement of the instruments can be updated as better estimates of PM2.5 concentrations become available after the initial placement of sensors.

such that at least a sensor or a monitor is placed at grid $a$). For each grid $a \in \{1, 2 \dots, n\}$, let $x_a$ be equal to one, if a sensor is

placed at grid $a$ otherwise it is equal to zero, $y_a$ be equal to one if a monitor is placed at grid $a$, otherwise it is equal to zero

and $z_a$ be equal to one if any instrument is placed at grid $a$, otherwise it is equal to zero. Let $c$ be the cost of a sensor, $c'$ be the

cost of a monitor and $P$ be the total available budget. Let $B$ be the set of grids where at least one sensor should be placed. Let

$C$ be the set of grids where a monitor cannot be placed. Let $h$ be the minimum number of monitors that should be deployed.

Let $M$ be a very large positive number and $m$ be a very small positive number. The formulation for optimally placing hybrid

instruments is as follows:

$$\text{Max } \sum_{a=1}^{n} m_a . g(d(a)) \tag{2}$$

$$\text{s.t. } \sum_{a=1}^{n} (cx_a + c'y_a) \leq P \tag{3}$$

$$\sum_{a \in B} x_a \geq 1 \tag{4}$$

$$\sum_{a \in C} y_a = 0 \tag{5}$$

$$\sum_{a=1}^{n} y_a \geq h \tag{6}$$

$$Mz_a + m \geq x_a + y_a , \forall a = 1,2,\dots,n \tag{7}$$

$$x_a + y_a \geq z_a , \forall a = 1,2,\dots,n \tag{8}$$

where $d(a) = \min_{b \, \epsilon \, V} \{z_b . d(a,b) + \overline{d}(a).(1 - z_b)\}$ and $\overline{d}(a) = \max_{b \, \epsilon \, V} d(a,b)$.

The objective is to choose a subset of grids $S \subseteq V$ that maximizes the overall satisfaction percentage under given constraints.

Here, we define $d(a,b)$ as the distance between grid $a$ and grid $b$ (note that when we are finding distances between two grids

we mean distances between the centres of the grids), $d(a)$ is the minimal distance between grid $a$ and any grid of set $S$

(assuming that $S$ is not an empty set, which is the case because of the constraint in Equation (4)). The condition in Equation

(3) is the budget constraint which states that the total cost of all instruments cannot exceed $P$. The condition in Equation (4)

ensures that a sensor is placed in at least one of the grids belonging to the set $B$. We do not put analogous constraints such as

Equation (4) for monitors as monitors cannot be place anywhere since they need where electricity availability, they are big,

heavy and costly as compared to sensors. Equation (5) ensures that no monitor is placed at any grid belonging to the set $C$

(these grids can belong to locations like open areas, areas near waterbodies, etc.). Note that it may not be cost-effective or

practical to deploy expensive monitors in certain areas and thus monitor deployments are restricted, but sensor deployments

are not. The condition in Equation (6) ensures that at least $h$ monitors are deployed. Equations (7) and (8) are the definitional

constraints for variable $z_a$. That is, they ensure that for each grid $a$, $z_a$ is equal to one if $x_a + y_a \geq 1$ otherwise, $z_a$ is equal

to zero.

As mentioned before, users may not be either interested or be able distinguish between the data collected from monitors and

sensors. However, the network designer may be interested in distinguishing between the satisfaction obtained from monitors

and sensors. Therefore, we provide an alternate optimization formulation that distinguishes between the satisfaction obtained from monitors and sensors in Appendix B.

## 2.2 Methods

We will now present different algorithms to solve the proposed formulation. We will first introduce Genetic Algorithm (GA).

### 2.2.1 Genetic Algorithm

A Genetic Algorithm is a metaheuristic that is inspired by the natural selection process and genetics (Deb, 2001). It mimics the principles of survival of the fittest, crossover, and mutation to iteratively search for optimal solutions. The algorithm starts by creating an initial population of potential solutions, represented as strings of individuals. Consider a string comprising of $2n$ elements ($n$ is the total number of grids), with the first $n$ elements are for the placement of sensors and the next $n$ elements are for the placement of monitors. Each element in the string can take a value of either 0 or 1, where 1 indicates the presence

of a sensor or monitor (depending on whether we are looking in the first $n$ or last $n$ elements) in the corresponding grid, and 0 indicates the absence. We now consider a modification of the above string where we remove the elements that correspond to monitors belonging to set $C$. The removed elements will always have value equal to zero due to the definition of set $C$ (consequently, monitors will not be placed on the grids belonging to the $C$ set) and thus they are separated so that the values of these elements do not change due to different processes in GA. The aforementioned modified string is used in our problem.

Each string encodes a set of decision variables, representing a candidate solution to the problem.

We define a fitness metric that is used to assign a relative merit (fitness) to each solution based on the corresponding objective function value and constraint violations. The fitness, $F(H)$, of any string $H$ is calculated as follows:

$$F(H) = \begin{cases} fn & \text{if } H \text{ is a feasible sol string} \\ fn_{min} - D_1 - D_2 - D_3 & \text{otherwise} \end{cases} \tag{9}$$

$$\text{Where, } D_1 = \begin{cases} 0 & \sum_{a=1}^{n}(cx_a + c'y_a) \leq P \\ \sum_{a=1}^{n}(cx_a + c'y_a) - P & \text{Otherwise} \end{cases} \tag{10}$$

$$D_2 = \begin{cases} 0 & \sum_{a \in B} x_a \geq 1 \\ 1 & \text{Otherwise} \end{cases} \tag{11}$$

$$D_3 = \begin{cases} 0 & \sum_{a=1}^{n} y_a \geq h \\ h - \sum_{a=1}^{n} y_a & \text{Otherwise} \end{cases} \tag{12}$$


Here, $fn$ is the objective function value for string $H$ as obtained by Equation (2), $fn_{min}$ is the minimum value of objective function values over all the feasible solution strings in a given population of strings, and $D_1$, $D_2$ and $D_3$ are penalty values for violating constraints in Equation (3), (4) and (6), respectively. Note that there is no penalty value for violating the constraint

in Equation (5) as that is automatically satisfied due to the way we define our strings (recall that we removed the elements corresponding to the grids of set $C$).

In each generation (or iteration) of GA, the Roulette Wheel Selection (RWS) is used to select solutions from a population based on their fitness values (Deb, 2001). RWS provides a proportional selection mechanism where fitter solutions have a higher probability of being selected, but it still allows weaker solutions to have some chance of being chosen. After the selection procedure, crossover procedure is followed where two strings are randomly selected from the mating pool, and a partial interchange from both strings is done to generate two new strings. We use the two-point crossover operator where two distinct crossover points divide the strings into three substrings and the middle substring is exchanged between the strings (Deb, 2001). After crossover, mutation procedure is carried out where the mutation operator alters 1 to 0 or vice versa in each element of a string with probability $P_m$ (referred to as the mutation probability). Note that mutation helps in maintaining diversity in the population. After applying the genetic operators, parent population and offspring population are combined, strings in the combined population are sorted in non-increasing order and the top half of the combined population is selected as the population for the next generation. This process is repeated over multiple iterations or generations until the termination criteria (to be specified next) is met. We now describe the termination criteria. Let the average fitness value of strings in the population of $i$th iteration or generation be $k_i$. Let $N$ be the maximum number of iterations of GA that are allowed. Then, the algorithm stops at the end of the $i$th iteration if $\left|\frac{k_i - k_{i-1}}{k_{i-1}}\right| \leq \alpha$ (where $\alpha$ is a given value) or if $i$ becomes equal to $N$.

### 2.2.2 Greedy Algorithm

The second method to solve the optimization problem from Section 2.1 is a Greedy Algorithm (GrA). A greedy algorithm iteratively comes up with a solution by making choices that are locally optimal in each iteration but it is not guaranteed to produce an optimal solution. In this algorithm, we first place a sensor at one of the locations from set $B$ to satisfy Equation (4). This placement is done by selecting the grid with the highest $m_a$ among the set $B$. Then, we find the placement location for $h$ monitors to satisfy Equation (6) by ensuring that Equation (5) (which tells us about the grids where monitors can't be placed) is not violated. We now define grid location $s^*$ with largest information gain as $s^* = argmax_s \sum_{a=1}^{n} m_a \left( g\left(d'(a, K \cup s)\right) - g\left(d'(a, K)\right)\right)$ where $K$ is the set of grids that have either a sensor or a monitor already placed (note that $K$ is not an empty set because we have at least one grid belonging to set $B$ that has a sensor placed) and $d'(a, K)$ represents the minimum distance between grid $a$ and any grid of set $K$. The placement of $h$ monitors is done by repeatedly choosing the grid location with the largest information gain $s^*$. Let $P' = P$, where $P'$ is the budget that remains after we subtract the cost of different instruments that are placed in different iterations of GrA. After the placement of one sensor plus $h$ monitors, the available budget $P' = P - c - hc'$. After satisfying Equation (6), there is no benefit of placing more monitors that are costly and thus we target to

place sensors. We keep placing sensors such that the grid location with the largest information gain $s^*$ is selected while ensuring that $P'$ is updated with every placement of sensor and budget constraint is satisfied. The algorithm terminates when there is an insufficient budget to place sensors, i.e., when $P' < c$.

We now provide an example to illustrate the greedy algorithm in Appendix C.

## 3 Results

In this section, we will present results by testing our proposed algorithms in different settings. Our algorithms have been employed in two distinct areas within Surat and Mumbai cities. Both algorithms were implemented in MATLAB and executed on a computer with Intel® Core™ i7-2600 processor and 8 GB RAM.

### 3.1 Surat City

We first consider a portion of Surat which is a major city in the state of Gujarat, India, for optimal placement of air quality instruments. In this study, we take a pilot project area of 5 km x 5 km in Surat and divide it into 25 grids (thus each grid is of the size 1 km x 1 km). The total number of grids in Surat are 25 which are numbered from 1 to 25 from left to right in the increasing order and from top to bottom in the increasing order (see Table 1). For calculating the optimal locations for hybrid instruments, we use the average percentage of population density (WorldPop provides open source[3] population density data at a spatial resolution of 1 km x 1 km) and $PM_{2.5}$ emission data (The Energy and Resources Institute (TERI) provided us $PM_{2.5}$ emission data for Surat city at a spatial resolution of 1 km x 1 km) for the part of Surat city that we focus. Figures 1 and 2 provide the intensity of population density (in population per sq. km) and emissions (in kT/yr) for the grids for Surat City that are considered in this paper.

**Table 1. Numbering of grids in the portion of Surat City that are considered.**

| 1 | 2 | 3 | 4 | 5 |
|----|----|----|----|----|
| 6 | 7 | 8 | 9 | 10 |
| 11 | 12 | 13 | 14 | 15 |
| 16 | 17 | 18 | 19 | 20 |
| 21 | 22 | 23 | 24 | 25 |

---

[3] https://hub.worldpop.org/geodata/summary?id=41746

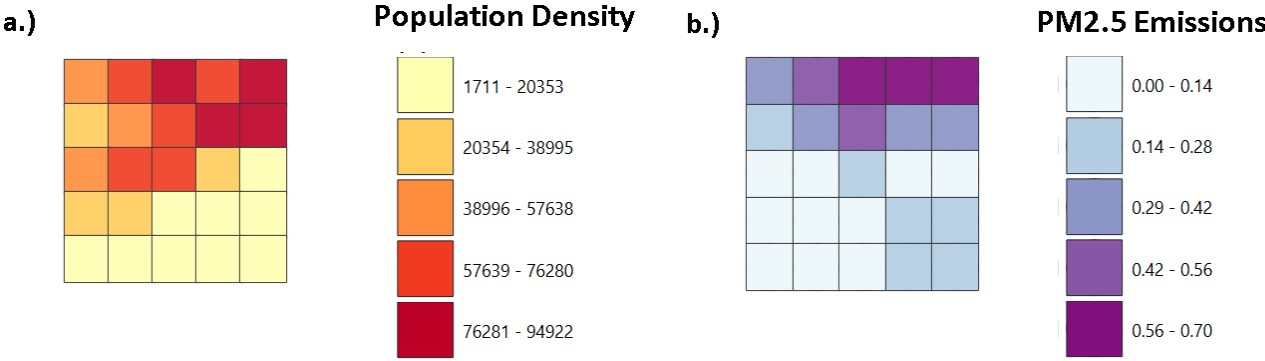

a.) Population Density

| | |
|---|---|
| | 1711 - 20353 |
| | 20354 - 38995 |
| | 38996 - 57638 |
| | 57639 - 76280 |
| | 76281 - 94922 |

b.) PM2.5 Emissions

| | |
|---|---|
| | 0.00 - 0.14 |
| | 0.14 - 0.28 |
| | 0.29 - 0.42 |
| | 0.42 - 0.56 |
| | 0.56 - 0.70 |


**Fig 1. Intensities of population density for the considered grids of Surat City (in population per sq. km) in the left and PM2.5 emissions data for the considered grids of Surat City (in kT/yr) in the right.**

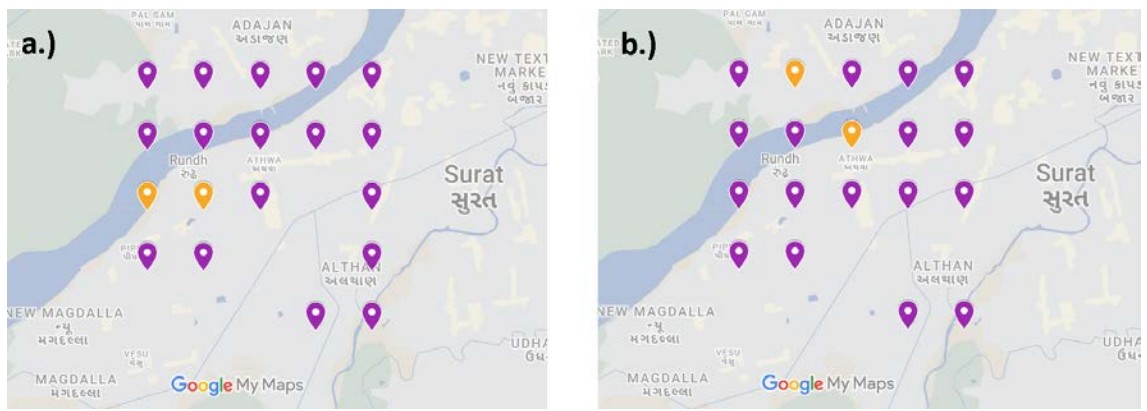


**Fig 2. Hybrid placement obtained by GA (left) and GrA (right) for the Surat network with budget value of $295000. Map data © 2023 Google.**

Figure 2 displays the placement locations of sensors (purple points) and monitors (orange points) in Surat city as obtained by
Genetic algorithm (left) and Greedy algorithm (right) with the budget value of $295000. Note that the objective function value corresponding to both the algorithms for this case is around 96.3 (see Figure 3) but the spatial distribution of the instruments is not the same. That is because this is a discrete optimization problem and it is possible that two solutions with very different looking spatial distribution can have the same objective function value. Note that there is no scope to further add any instrument in the solution of any of the algorithms as there are two monitors and seventeen sensors and 2*122000 + 17*3000 = 295000.
Also, note that the weights taken in the objective function are $w_1 = w_2 = 0.5$. That is because by averaging these variables, we strike a balance between the need to monitor areas with high pollution levels (captured by $PM_{2.5}$ emissions) and areas with

high population density (captured by the population density). Note that we will present the sensitivity analysis with different weights later. The parameter values that are used in this placement are as follows: cost of a sensor ($c$) is \$3000, cost of a monitor ($c'$) is \$122000,[4] total available budget ($P$) is \$295000, value of $\theta$ and $h$ are 1 and 2, respectively. The GA parameters that are used are as follows: population size is equal to 1000, mutation probability ($P_m$) is equal to 0.1, maximum number of iterations or generations is 500 and value of $\alpha$ is $10^{-5}$. Note that we determined that $\alpha = 10^{-5}$ consistently yielded satisfactory convergence while ensuring computational efficiency through systematic tuning involving a range of $\alpha$ values.

Figure 3 shows the values obtained and computational time for the two algorithms, considering different total available budgets (i.e., $P$). Note that the obtained value on the vertical axis in Figure 3 is objective function value as given by Equation (2). The minimum budget that is considered is \$253,000, which is equal to the cost of three sensors plus $h$ monitors (any value of budget lower than this will not yield a feasible solution of the problem as the budget constraint will not get satisfied). The maximum budget in Figure 3 is \$313,000, which allows for the placement of 2 monitors and 23 sensors, covering the entire portion area (as there are a total of 25 grids) under minimum possible budget as at least 2 monitors need to be placed by Equation (6). Note that if the budget is sufficiently large and the optimal solution involves covering all the grids then GA can provide solutions where at some places interchanging sensors with monitors will not change the value of the solution. That is because the objective function does not differentiate between monitors and sensors and the solutions of GA are generated through a probabilistic process and thus may exhibit a different spatial distribution than that obtained by GrA.

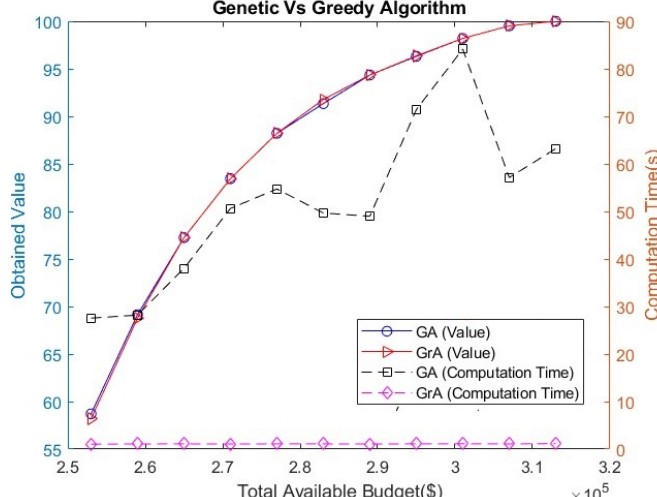

**Fig 3. Plot comparing genetic vs greedy algorithms for varying total available budget values.**

---

[4] We obtained the cost estimate for a monitor through the cost of continuous ambient air quality monitoring stations (CAAQMS) imported to India whose price is available at the following link: https://timesofindia.indiatimes.com/india/centre-asks-states-not-to-procure-imported-air-quality-monitors-indigenous-systems-to-be-deployed/articleshow/95901936.cms. Similarly, the cost of a sensor (here Aeroqual S500) is estimated from the following link: https://www.cleanair.com/product/aeroqual-s500-starter-kit/.

From Figure 3, it can be observed that, for most budget points, the obtained values for GrA and GA are very close. Also, note that the obtained values for both the algorithms increase with the increase in budget because it is possible to place more instruments with the increase in budget and that results in increase in the overall satisfaction function value. Note that the computation time of GA is significantly larger than that of GrA because GA samples through a set of possible solutions and iteratively applies various operators such as selection, crossover and mutation whereas GrA is a deterministic algorithm that comes up with a single solution.

We provide an example in Appendix D to show the performance of different network configurations that have different variations with respect to the optimal solution.

### 3.1.1 Sensitivity Analysis

In this section, we will present the results with a different $g(d)$ function and consider different weighs corresponding to $p_a$ and $e_a$ in the objective function.

### 3.1.1.1 Sensitivity analysis with another $g(d)$ function

As previously mentioned, the $g(d)$ function should be a decreasing function. Therefore, we explore an alternative function $g(d) = \frac{1}{d+1}$ apart from the exponential function. We have now obtained the results by greedy algorithm and genetic algorithm for Surat city network (5x5 size) using $g(d) = \frac{1}{d+1}$, while keeping all the other parameters the same (as in Figure 3).

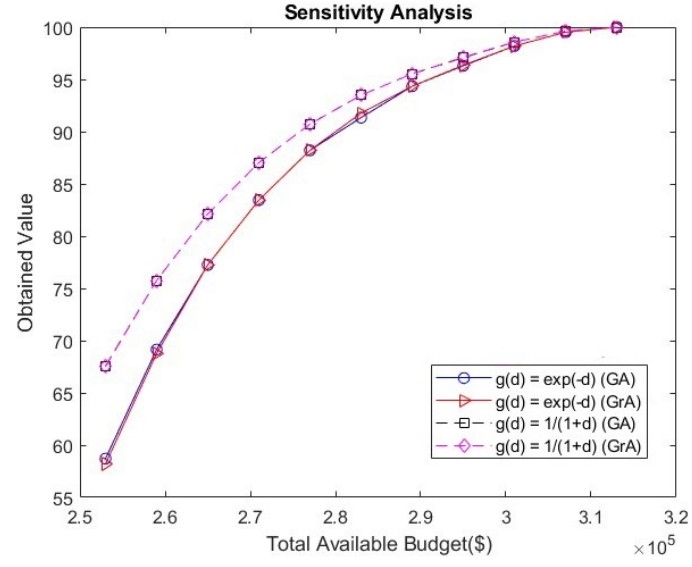

**Fig 4. Plot comparing two functional forms of $g(d)$.**

Figure 4 presents the values obtained by different algorithms and functional form for $g(d)$ with varying budget values. It can be seen in the above figure that the values obtained by the genetic algorithm and the greedy algorithm for $g(d) = e^{-d}$ are very close and thus the solid blue and red curves almost overlap. The same holds for $g(d) = \frac{1}{d+1}$ and therefore the dashed black and purple lines almost overlap. Note that the values that are obtained by the two algorithms for $g(d) = \frac{1}{d+1}$ are greater than

that obtained for $g(d) = e^{-d}$. That is because $\frac{1}{d+1} > e^{-d}$ for all positive values of $d$. However, notice that the pattern of the values that are obtained for the two functional forms is the same, i.e., the values decreases as the total available budget increases. Also, notice that the values obtained by the two functional forms converge at the budget value of \$313,000. Since it is not possible to have percentage values greater than 100, the values for both the functional forms will remain the same for budget values greater than \$313,000. We believe that similar patterns will be observed by other functional forms of $g(d)$ as

long as they satisfy the conditions that are necessary for satisfaction functions (i.e., $g(d)$ must be a decreasing function and $g(0) = 1$). We have defined a similarity index that quantifies the difference in the placement of hybrid instruments as obtained by different algorithms. Suppose the number of grids where the placement of hybrid instruments by the two algorithms is identical is given by $k$ (a grid is said to have identical placement by the two algorithms if the grid contains a sensor as determined by both the algorithms or a monitor as determined by both the algorithms). Also, let the maximum number of

hybrid instruments that can be placed in the given constraints be equal to $p$. Then, similarity index is given by $k/p$. Since the solution obtained by the genetic algorithm is probabilistic, we tested five runs of genetic algorithm (for a given budget value) and compared the solution obtained by each run to the solution obtained by the greedy algorithm to determine the similarity indices and finally obtained the average similarity index by taking the mean of five similarity indices. Figure 5 shows the average similarity index for different budget values and for different $g(d)$ functions (while keeping equal weights for the

percentages of population density and emissions). Note that similarity index is upper bounded by one. Also, we see that as budget values increase the average similarity index for both $g(d)$ functions increase. That is because as the budget increases the number of grids at which instruments can be placed increases and both the algorithms usually place sensors at most grids except at a few grids where monitors are placed to meet the requirement of minimum monitors. Note that the average similarity index is around 0.5 for low budget values due to the existence of solutions that have varying placements but have close

objective function values (but as the budget increases the variation in the placement reduces as explained before). Also, the average similarity indices obtained by the two $g(d)$ functions are close for most of the budget values.

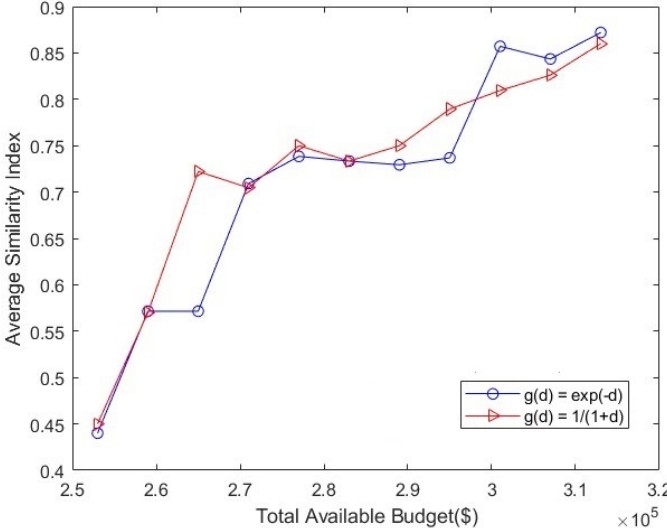

**Fig. 5 Average similarity index for different $g(d)$ functions.**


### 3.1.1.2 Sensitivity Analysis for different weights in the objective function

We have also conducted the sensitivity analysis by varying the weights between the percentages of population density and PM2.5 emissions (i.e., $p_a$ and $e_a$) for Surat city. Table 2 shows the weights corresponding to the different cases that have been considered. We have determined the results for both Greedy algorithm and Genetic algorithm by keeping all the parameters

same (as in Figure 3).

**Table 2. Different cases for the weights**

| Case | Weightage for $p_a$ | Weightage for $e_a$ |
|------|---------------------|---------------------|
| 1 | 0.25 | 0.75 |
| 2 | 0.5 | 0.5 |
| 3 | 0.75 | 0.25 |

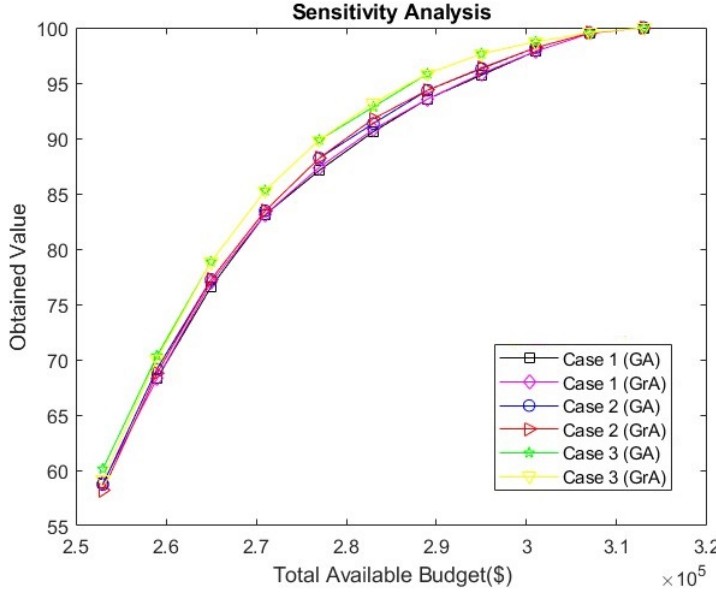

 **Fig 6. Plot comparing cases for different weights corresponding to $p_a$ and $e_a$.**

Figure 6 shows the values that are obtained for different cases, budget values and algorithms. As before, the values obtained by GA and GrA are very close for given weights and budget. Among these cases, the values corresponding to Case 3 (where $p_a = 0.75$ and $e_a = 0.25$) are the highest and that corresponding to Case 1 (where $p_a = 0.25$ and $e_a = 0.75$) are the lowest.

Thus, as the relative weightage for population density increases in the objective function, the values obtained increases. However, it can be seen that the difference between the values for Cases 1 and 3 is not that large, signifying that the objective function values may not be that sensitive to the relative weightage between population density and emissions. Figure 7 shows the average similarity index for different budget values and different cases corresponding to the weights of percentages of population density and PM2.5 emissions (while keeping $g(d)$ as exponential function). It can be seen that the average

similarity index increases with budget values for the same reason as mentioned for Figure 5. Also, the values of similarity indices are close for most of the budget values.

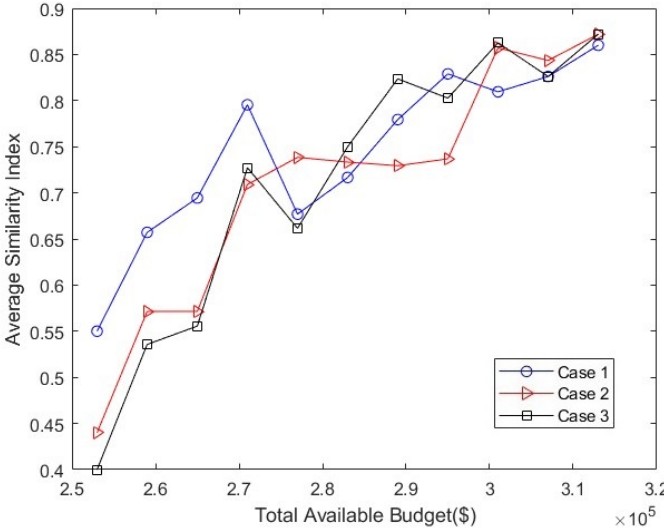

Fig. 7 Average similarity index for different weights corresponding to $p_a$ and $e_a$.


## 3.2 Mumbai City

We now present the results that we tested for portions of Mumbai, which is the financial hub of India. In this case, we only considered the contribution of population in the objective function (i.e., $w_1 = 1, w_2 = 0$, implying $m_a = p_a$) due to unavailability of $PM_{2.5}$ emission data for Mumbai city. However, the aforementioned change does not have any significant

issue on the results that we present as we plan to test the effect of varying the budget (as in the last section) and the effect of varying the size of the network (i.e., the number of grids). All the parameter values for the algorithm's execution were the same as in the example for Surat city (i.e., Figure 3), except for the variable $\theta$, which has now been set to 5 (note that $\theta$ has been increased now because we have a larger number of grids in Mumbai network as compared to Surat, resulting in higher average distances between the grids for the Mumbai network and thus we need to update $\theta$ for better normalization). Consider a region

of size 10 km x 10 km in Mumbai City that has been divided into 100 grids (i.e., each grid is of the size 1 km x 1 km). Figure 8 shows the variation of values obtained and computation time with total available budget for GA and GrA for this region. The solid lines represent the obtained values and dashed lines are used to represent the computation time in seconds for different algorithms. It can be seen that the genetic algorithm (GA) provides higher value as compared to the greedy algorithm (GrA) for most of the cases. Thus, it highlights the importance of GA in obtaining values that are closer to the optimal as compared

to GrA when the network size increased (however this advantage comes at the high computational cost of GA as compared to GrA).

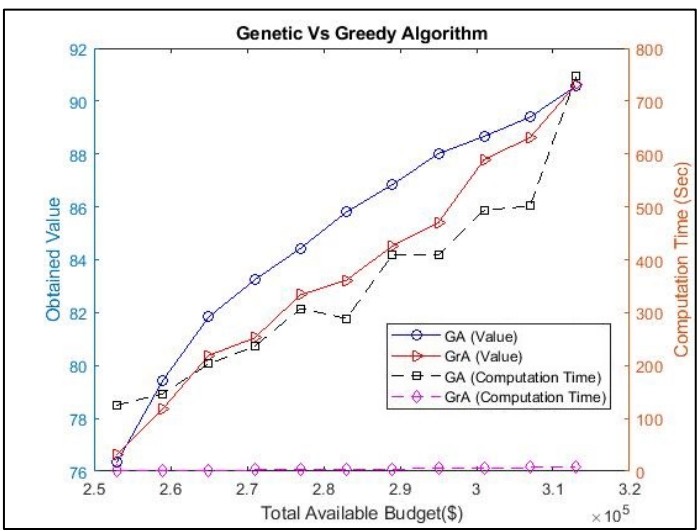

**Fig 8. Plot comparing genetic versus greedy algorithm for varying total available budget values.**


Figure 9 shows the placement of hybrid instruments obtained for the two algorithms (GA and GrA) when the budget is equal to $283000 when we have all the other parameters the same as that in Figure 8. The blue and orange points represent the placement of sensors and monitors, respectively. In Figure 9a, two sensors are positioned in the northeast area, while no sensors or monitors are placed in that area in Figure 9b. In Figure 9b, monitors/sensors are predominantly concentrated on the left side

of the Mumbai area, whereas in Figure 9a, the sensors/monitors exhibit a more diverse and scattered distribution. Note that out of 100 grids, sensors and monitors can be placed in only 15 grids by maximizing the objective function. The leftmost and southern areas have the highest population density, which explains the concentration of sensors and monitors in those regions. There is difference in the solutions that are obtained by the two algorithms because GA samples through various solutions that to proceed towards a solution is closer to the optimal whereas GrA is a deterministic algorithm and may get stuck near a locally

optimal solution.

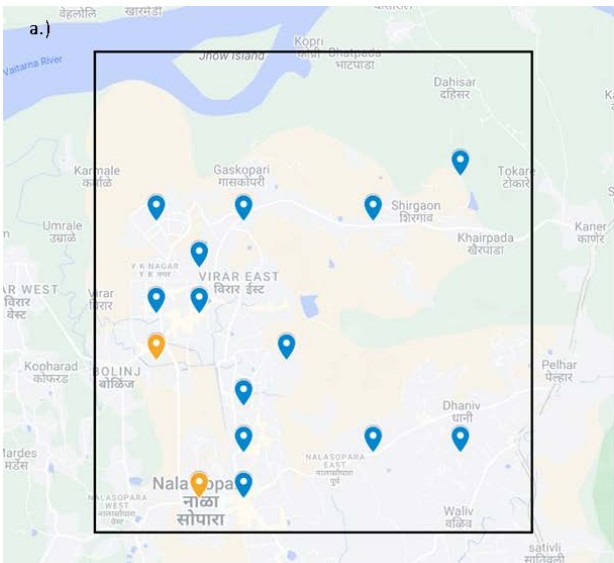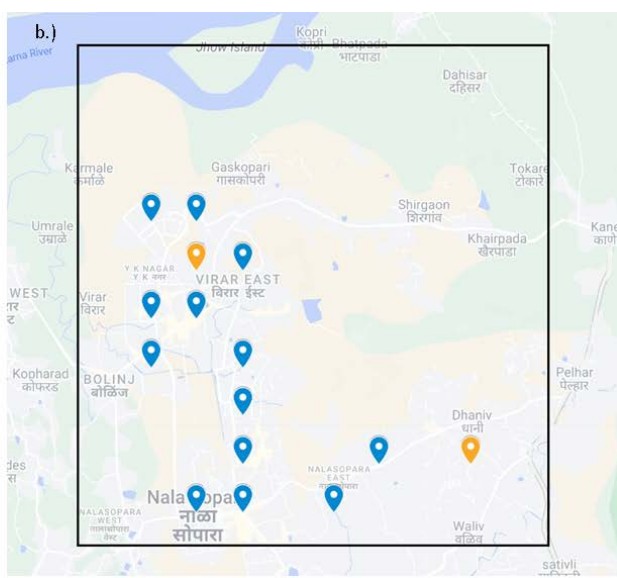

**Fig 9. Sensor placement obtained by GA (left) and GrA (right) for 10 km x 10 km (100 grids) region in Mumbai when the budget is equal to $283000. Map data © 2023 Google.**


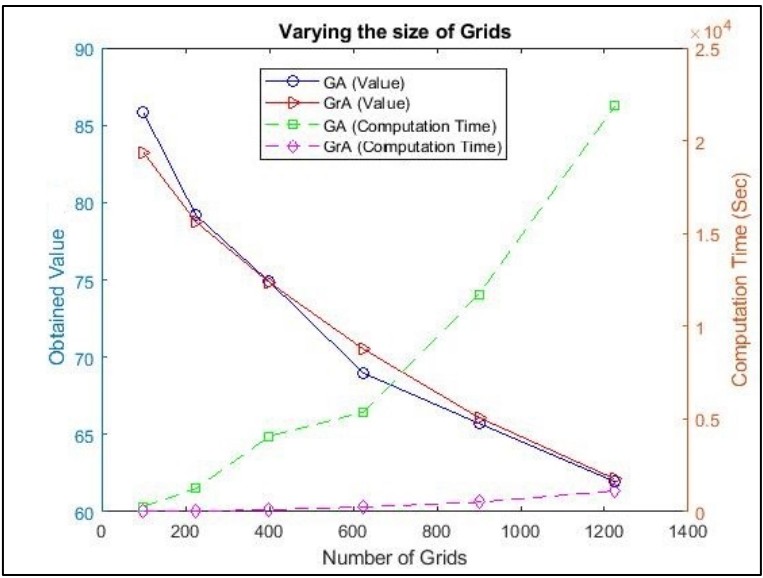

**Fig 10. Plot comparing genetic and greedy algorithms for varying number of grids.**

Figure 10 shows the comparison between GA and GrA with varying number of grids for the budget value of \$283000.[5] The solid lines represent the obtained values in percentage for different algorithms and dashed lines are used to represent the computation time in seconds for different algorithms. As the number of grids increases, there is a noticeable decline in citizen satisfaction (i.e., the obtained values) because the budget $P$ remains the same and thus the satisfaction averaged across all the grids reduces as it gets distributed across the total region (note that the percentage of population in each grid also reduces as

the number of grids increase and thus that also contributed to the observed trend). Also, the values obtained by GA and GrA are similar and in some cases, GA outperforms GrA whereas the reverse happens in other cases. Note that the computation time required for GA increases rapidly with the increase in the number of grids because with the increase in the number of grids, the size of each string in GA increases and it takes more iterations before the termination criterion is reached in GA (as the number of feasible solutions increase with the increase in grid size). However, the increase in the computational time of

GrA is not that high as it is a polynomial-time algorithm (Cormen et al., 2022), i.e., the computational time increases polynomially with respect to the increase in the problem size (i.e., the number of grids in our problem).

## 4 Conclusions

This research paper proposed an optimization formulation for placement of hybrid instruments (sensors and monitors). The objective of the problem is to maximize the satisfaction function while satisfying various constraints for the placement. To

solve this formulation, we proposed two algorithms: a genetic algorithm (GA) which is a metaheuristic that works using the principles of evolution and a greedy algorithm (GrA) that makes choices that are locally optimal in each iteration. We tested the placement solutions generated by these algorithms on networks from different locations (Surat and Mumbai) that differed over sizes and characteristics (population distribution, budget and $PM_{2.5}$ distribution). We observed that as the total available budget increased, the obtained values from the two algorithms also increased as it became possible to place more instruments

(sensors and monitors). We found that GrA is very computationally efficient as compared to GA, but we found that both GrA and GA provided close values (in some cases GA outperformed GrA whereas in other cases the reverse happened). Note that since GA searches through a set of solutions over multiple iterations and uses operators like mutation it has a better likelihood of getting towards the optimal solutions whereas GrA may get stuck near a local optimum in some cases. These findings suggest that if time is not constrained (i.e., we have a few days to decide the placement solution) it might be better to use GA

and GrA together (i.e., use the best solution out of the two algorithms) to place the instruments whereas in scenarios where there is scarcity of time, it is advised to use GrA. While the study's results are specific to these locations, the underlying methodology and principles learned from these cases can be broadly applied to other areas facing similar air quality monitoring challenges. The methodology presented in our paper serves as a template for optimizing sensor networks in any location,

---

[5]    The population data (in terms of population per square km) is available at the following link: https://docs.google.com/spreadsheets/d/1tdDUXnu4EQb2t3g_M96RXb4mkRXdaFP3/edit#gid=1468141414 . This data contains the largest set of grids used with 35 x 35 = 1225 grids. There are two sheets, one shows the numbering of grids and the other contains the population data. The population data for both Surat and Mumbai have been obtained from the following website: https://hub.worldpop.org/geodata/summary?id=41746

provided that relevant data on population, emissions, and potential grid locations are available. Our research aims to provide valuable insights for future government decision-making processes regarding the optimal deployment of hybrid instruments in cities lacking an existing sensor network.

There are several interesting future extensions of this work that are possible. We acknowledge the challenges associated with quantifying PM2.5 emissions in areas lacking an established monitoring network, as evident in the Mumbai case. However, in future, solutions such as considering existing models or satellite-derived data as proxies for local PM2.5 concentrations during the network design phase can be implemented. Also, after the placement of instruments, one could iteratively update the placement of the network using some existing models or proxy data as newly collected data update the prior estimates of concentrations in the different grid cells. In addition, we assumed a particular form of the satisfaction function (consisting of exponential terms) but other forms can also be tested. Similarly, other factors apart from population density and $PM_{2.5}$ concentrations such as socio-economic disparities across various grids can also be factored while determining the satisfaction function. Note that exploring other objective functions such as improving estimates of population exposure, monitoring the largest known sources, etc., would also be very interesting. To address these alternative objectives, we could make the following modifications to our approach. First, the objective function could be defined appropriately whether it is optimizing public satisfaction, estimating exposure, or addressing specific environmental issues. Also, depending on the chosen objective, we may need to adapt the data collection methods used. For example, if the goal is to estimate population exposure, we may need to tailor the data collection frequency accordingly. The analysis methods and models used for decision-making can be customized based on the objective. For instance, if the goal is to address specific environmental concerns, sophisticated modelling techniques may be employed to assess pollutant dispersion. When other objective functions are used then the fitness function in the genetic algorithm will get modified. The selection, crossover and mutation operators will not change if the constraints remain the same and there would only be change in the objective function. Similarly, the greedy algorithm will have a modified gain function $s^*$ and the rest of the algorithm will remain the same provided the constraints in the problem remain the same. Thus, our approach can be flexibly adapted to address a range of objectives. Note that there is also potential for creating user-friendly software tools or decision support systems based on the methodology presented in our paper. Such tools would enable users with limited algorithmic expertise to apply similar optimization techniques to their specific locations, addressing the concern of not having the ability to run the algorithm. In these software tools, the users will only have to provide input values for the problem like the network they want to solve, costs of instruments, budget, the algorithm they want to use, etc., and the toolbox will provide the results.

## Appendix A

**Table A1**

| Notations | Description |
|:---:|:---:|
| $V$ | Set of all grids |
| $n$ | Total number of grids |
| $S$ | Set of grids selected for deploying hybrid instruments |
| $g(d)$ | An individual's satisfaction as a function of his or her distance $d$ to the closest sensor or monitor |
| $\theta$ | Exponential decay parameter |
| $p_a$ | Percentage of population living in grid $a$ |
| $e_a$ | Percentage of concentration of $PM_{2.5}$ in grid $a$ |
| $m_a$ | Weighted average of $p_a$ and $e_a$ |
| $c$ | Cost of each sensor |
| $c'$ | Cost of each monitor |
| $P$ | Total available budget |
| $h$ | Minimum number of monitors to be deployed |
| $z_a$ | Binary variable signifying whether a sensor or a monitor is placed at grid $a$ or not |
| $x_a$ | Binary variable signifying whether a sensor is placed at grid $a$ or not |
| $y_a$ | Binary variable signifying whether a monitor is placed at grid $a$ or not |
| $B$ | Set of grids where at least one sensor is to be placed |
| $C$ | Set of grids where monitors cannot be placed |
| $M$ | A very large positive number |
| $m$ | A very small positive number |
| $P_m$ | Mutation probability |
| $N$ | Maximum number of iterations of GA that are allowed |
| $d(a)$ | Minimum distance between grid $a$ and the grids containing hybrid instruments |
| $d(a,b)$ | Distance between grid $a$ and grid $b$ |
| $\overline{d}(a)$ | Maximum distance between grid $a$ and any other grid of set $V$ |
| $d'(a,K)$ | Minimum distance between grid $a$ and set $K$ |

## Appendix B

We provide an alternate optimization formulation whose objective is to maximize the weighted sum of satisfaction functions from monitors and sensors. Let $w_s$ be the weight corresponding to the satisfaction from sensors and $w_m$ be the weight corresponding to the satisfaction from monitors. Let $d(a)$ be the minimum distance between grid $a$ and any grid containing

sensors and $d'(a)$ be the minimum distance between grid $a$ and any grid containing monitors. The remaining parameters and variables mean the same as before. Then, the formulation is as follows:

$$\max w_s \sum_{a=1}^n m_a \cdot g(d(a)) + w_m \sum_{a=1}^n m_a \cdot g(d'(a)) \qquad (B1)$$

$$\text{s.t. } \sum_{a=1}^n (cx_a + c'y_a) \leq P \qquad (B2)$$

$$\sum_{a \in B} x_a \geq 1 \qquad (B3)$$

$$\sum_{a \in C} y_a = 0 \qquad (B4)$$

$$\sum_{a=1}^n y_a \geq h \qquad (B5)$$

where $d(a) = \min_{b \in V} \{x_b \cdot d(a,b) + \overline{d}(a) \cdot (1 - x_b)\}$,

$\overline{d}(a) = \max_{b \in V} d(a,b)$ and $d'(a) = \min_{b \in V} \{y_b \cdot d(a,b) + \overline{d}(a) \cdot (1 - y_b)\}$.

Thus, the relative values of the weights $w_s$ and $w_m$ decide the relative importance being given to monitors and sensors. Typically, $w_m$ should be chosen larger than $w_s$ as monitors are more accurate than sensors. One could solve the above formulation with minimal changes to the proposed genetic and greedy algorithms.

## Appendix C

We now provide an example of a 3x3 network (i.e., a network having 3x3 = 9 grids) to illustrate the greedy algorithm. The population density data (in population per sq. km) and PM2.5 emissions data (in kT/yr) for a 3x3 network are provided below on the left and right, respectively.

**Table C1: Population density data**

| | | |
|---|---|---|
| 65646 | 29660 | 15504 |
| 9487 | 2984 | 2260 |
| 2042 | 2393 | 1711 |

**Table C2: PM2.5 emissions data**

| | | |
|---|---|---|
| 0.143405 | 0.120589 | 0.097773 |
| 0.114025 | 0.142434 | 0.170843 |
| 0.084646 | 0.16428 | 0.243914 |

Then we calculate the percentage of population density $(p_a)$ and PM2.5 emissions $(e_a)$ for each grid and then calculate $m_a$ which is an average of $p_a$ and $e_a$. The following tables show the values of $p_a$ (left) and $e_a$ (right).

**Table C3: Percentage of population density**

| | | |
|---|---|---|
| 49.85 | 22.5231 | 11.7734 |
| 7.2042 | 2.266 | 1.7162 |
| 1.5506 | 1.8172 | 1.2993 |

**Table C4: Percentage of PM2.5 emissions**

| | | |
|---|---|---|
| 11.1868 | 9.407 | 7.6271 |
| 8.895 | 11.111 | 13.3273 |
| 6.6031 | 12.8153 | 19.0274 |

The following values are the $m_a$ values for each grid of the 3x3 network that we consider.

**Table C5: Average of the percentages of population density and PM2.5 emissions**

| | | |
|---|---|---|
| 30.5184 | 15.965 | 9.7002 |
| 8.0496 | 6.6885 | 7.5217 |
| 4.0769 | 7.3162 | 10.1633 |

Suppose the set $B$ in which at least one sensor is to be placed from Equation (4) is consists of grids 7 and 9 and set $C$ in which no monitor can be placed from Equation (5) is given by set 7. Suppose $h = 2$, which represents the minimum number of monitors required. Let the cost of sensor ($c$) and monitor ($c'$) be 200 and 8000 units respectively. The total available budget be 16500 units.

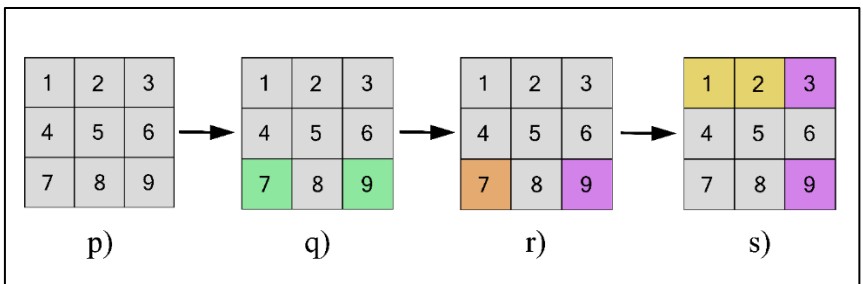

**Fig C1. Example to show the working of greedy algorithm**

Figure C1p shows the initial empty grids which are grey in color. Figure C1q shows the grid area in which two grids (i.e., grids 7 and 9) are shown in light green color grids which tells us about grids in set $B$, where at least one sensor must be placed. Given that the value of $m_a$ for grid 9 is greater than that of grid 7, a sensor is initially placed in grid 9 to satisfy Equation (4). The placement of a sensor at grid 9 reduces the available budget to 16300 units.

Figure C1r shows the placement of sensor at grid 9 and a grid (corresponding to set $C$) which is shown by orange colored square grid (i.e., grid 7). The monitors are positioned at grid 1 and 2 based on the values obtained from the largest information gain $s^*$ and in adherence to the Equation (5) which has a requirement that no monitor be placed on any grid belonging to set $C$. This further reduces the budget from 16300 units to 300 units by subtracting 16000 units (i.e., $c'h$)

We continue to place sensors until the budget constraint is violated. We will place next sensor at the grid with largest information gain $s^*$ and that grid is grid 3. This further reduces the budget from 300 units to 100 units. The algorithm stops here as there is no sufficient budget to proceed. Figure C1s shows the final solution using greedy algorithm where grey colored square grids show the empty grids, purple-colored square grids show the placement location of sensors and light yellow colored square grids shows the placement location of monitors.

**Appendix D**

We present an example to show the performance of different network configurations that have different variations with respect to the optimal solution. Consider an example of 3 x 3 network. Let the cost of a sensor ($c$) and a monitor ($c'$) be $3000 and $122000 respectively. Suppose the budget value is equal to $253000. The numbering of the grids follows the convention that numbers first increase as we go from left to right in the increasing order and numbers increase as we go from top to bottom. Let the set $B$ in which at least one sensor is to be placed from Equation (4) consist of grids 7 and 9 and set $C$ in which no

monitor can be placed from Equation (5) be set 7. We consider four different feasible solutions as follows:

Case 1: Solution obtained from greedy algorithm.

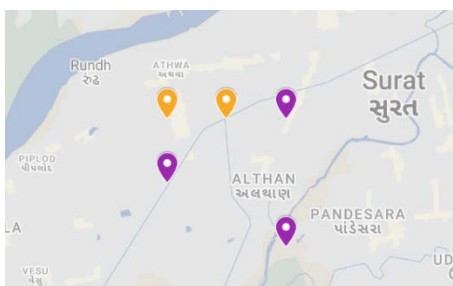

**Fig D1. Hybrid placement obtained by GrA. Map data © 2023 Google.**

Figure D1 shows the solution that is obtained in Case 1. The purple points show the placement location of sensors and orange points show the placement location of monitors. It can be seen that monitors are placed at grids 1 and 2 and sensors are placed at grids 3, 4 and 9.

Case 2: Sensor placed at grid 3 in Case 1 is moved to grid 7 (all the other instrument locations remain the same as in Case 1).

Case 3: Monitor placed at grid 1 in Case 1 is moved to grid 5 (all the other instrument locations remain the same as in Case 1).

Case 4: When sensor placed at grid 3 in Case 1 is moved to grid 7 and monitor placed at grid 1 in Case 1 is moved to grid 5 (all the other instrument locations remain the same as in Case 1).

The following table shows the values that are obtained for different cases.

| Case | Obtained Value |
|---|---|
| Case 1 | 83.8154 |
| Case 2 | 80.2608 |
| Case 3 | 68.7521 |
| Case 4 | 65.1975 |

It can be seen that Case 1 has the largest value and the value decreases as we go from Case 1 to Case 4. Thus, Case 1 is the closest to the optimal solution and Case 4 is the farthest. Note that Case 4 has both the modifications that are made in Cases 2 and 3 with respect to Case 1. Since there was a decrease in the value as we go from Case 1 to Case 2 and a decrease in value from Case 1 to Case 3, the largest decrease in value is seen as we go from Case 1 to Case 4.

## Author Contribution

HG and SNT led the conceptualization of this work. NA did the data curation. HG proposed the methodology. NA performed the coding and software part. HG and SNT supervised this work. NA prepared the original draft. All the authors contributed to review and editing.

## Competing Interests

The contact author has declared that none of the authors has any competing interests.

## Data availability

The data sets used in this study have been provided in the manuscript.

## Acknowledgements

The authors would also like to acknowledge the support of Centre of Excellence (ATMAN) approved by the office of the Principal Scientific Officer to the Government of India. The CoE is supported by philanthropies including Bloomberg Philanthropies, the Open Philanthropy and the Clean Air Fund and author SNT also acknowledges J C Bose award (JCB/2020/000044).

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
