# Peer review of "Hybrid Instruments Network Optimization for Air Quality Monitoring"

_Atmospheric Measurement Techniques, 2023_

## Author Comment (AC1)

**Response to Reviewer 1 Comments on "**Hybrid Instruments Network Optimization for Air Quality Monitoring**" (Manuscript** AMT-2023-173**)** submitted to *Atmospheric Measurement Techniques*

The original review comments are included in normal black text, and our responses to each comment follow in *italicized text*.

**Reviewer 1**

Overall the paper is well written and clearly presents its methodology and results.

I find that there is a conceptual problem with requiring emissions information for PM2.5 as an input to your optimization framework, when, in the absence of a monitoring network, this may not be well quantified (as is the case for the Mumbai example you present). You may want to discuss this as a potential limitation of your work and present some possible solutions, e.g., using existing model or satellite-derived information as a proxy for local concentrations during the network design phase, and/or iteratively chancing the network as newly collected data update the prior estimates of concentrations in the different grid cells. Furthermore, there is a potential disconnect between PM2.5 emissions and PM2.5 concentrations (which are to be measured by the network), with the possible impacts of secondary aerosol formation and pollution transport not being accounted for by using emissions information alone; maybe emissions are being used as a proxy for concentrations, but that was not clarified in the text.

*Response: We appreciate the reviewer's feedback. We acknowledge the challenges associated with quantifying PM2.5 emissions in areas lacking an established monitoring network, as evident in the Mumbai case. We have now included these solutions such as considering existing models or satellite-derived data as proxies for local PM2.5 concentrations during the network design phase. Also, after the placement of instruments using some existing models or proxy data, one could iteratively update the placement of the network as newly collected data update the prior estimates of concentrations in the different grid cells. We have now added this in the starting of the second paragraph in the Conclusions section (see page 19).*

*We agree with the distinction between PM2.5 emissions and PM2.5 concentrations. In our approach, we initially prioritize PM2.5 emissions as the foundational data for network design. Our primary objective is to strategically deploy sensors and monitors to identify and analyze*

*pollution sources in a given area. While we acknowledge the non-linear relationship between concentrations and emissions, the absence of concentration data at the grid level led us to utilize emissions as a proxy. However, the placement of the instruments can be updated as better estimates of PM2.5 concentrations become available after the initial placement of sensors. The above discussion has been added as a footnote on page 4.*

**Reviewer 1**

Similarly, while you clearly state that you are aiming to optimize public satisfaction through your sensing network design, there are many other potential objectives which might be the goal of a monitoring network, e.g., improving estimates of population exposure, monitoring the largest known sources, etc. I would suggest adding some commentary to your conclusions discussing how your approach might be modified to achieve these other objectives.

*Response: Thank you for your valuable feedback. We acknowledge that, in addition to optimizing public satisfaction, monitoring networks can serve various other important objectives. These objectives may include improving estimates of population exposure, monitoring the largest known sources of concern, addressing specific environmental or health concerns, or even optimizing resource allocation. To address these alternative objectives, we could make the following modifications to our approach. First, the objective function could be defined appropriately whether it is optimizing public satisfaction, estimating exposure, or addressing specific environmental issues. Also, depending on the chosen objective, we may need to adapt the data collection methods used. For example, if the goal is to estimate population exposure, we may need to tailor the data collection frequency accordingly.*

*The analysis methods and models used for decision-making can be customized based on the objective. For instance, if the goal is to address specific environmental concerns, sophisticated modeling techniques may be employed to assess pollutant dispersion.*

*When other objective functions are used then the fitness function in the genetic algorithm will get modified. The selection, crossover and mutation operators will not change if the constraints remain the same and there would only be change in the objective function. Similarly, the greedy algorithm will have a modified gain function $s^*$ and the rest of the algorithm will remain the same provided the constraints in the problem remain the same. Thus, our approach can be*

*flexibly adapted to address a range of objectives. The above stated discussion has been included in revised manuscript in lines 436-447 on pages 19 and 20.*

**Reviewer 1: General Comments:**

While you note that sensors and monitors have different capabilities, in your formulation they are treated equally in terms of your utility function (i.e., people will be equally satisfied to be located near a monitor or near a sensor). Could you justify this further, or discuss how your results might differ if monitors were given a higher weight?

*Response: We thank the reviewer for this comment. We accept that utility should not be the same for sensors and monitors if the accuracy of these instruments is known to the users and our optimization formulation can be modified to consider that. However, in many practical air quality monitoring scenarios, users may not be either interested or be able distinguish between data collected from monitors and sensors (if the information related to the type of instrument is not openly available), particularly in a hybrid network. From the user's perspective, the primary concern may be just to obtain reasonable air quality information, rather than worry about the specific source of the data. We have now added this lines 98-102 on page 4.*

**Reviewer 1: Specific Comments**

1. Line 10: Use of "reasonable" here is a bit unspecific; I suggest "less accurate" or a similar description instead, to contrast them with the reference stations.

*Response: We agree with the reviewer's comment and have replaced the term "reasonable" with "less accurate" in line 11 of page 1.*

2. Line 11: Remove "as"

*Response: We agree with the reviewer's comment and have removed the term "as" on line 11 of page 1.*

3. Line 17: "selects locally best choice" should be "selects the locally best choice"

*Response: We agree with the reviewer's comment and have replaced the term "selects locally best choice" with "selects the locally best choice" in line 18 of page 1.*

4. Line 27: Remove "the"

*Response:* *We agree with the reviewer's comment and have removed the term "the" in line 27 of page 1.*

5. Line 54: "limitations that" should be "limitations in that"

*Response:* *We agree with the reviewer's comment and have replaced the term "limitations that" with "limitations in that" in line 55 of page 2.*

6. Line 56: Suggest replacing "in the previous-to-previous paragraph" with "previously"

*Response:* *We agree with the reviewer's comment and have replaced the term "in the previous-to-previous paragraph" with "previously" in line 57 of page 2.*

7. Lines 60-61: The distinction between sensors and monitors has already been defined earlier in the paper

*Response:* *We agree with reviewer's comment and has removed this sentence "In this paper, we propose deploying a combination of low-cost sensors (referred to as sensors) and reference stations (referred to as monitors), termed hybrid instruments, in a specific region."*

8. Line 64: This definition for hybrid instruments has already been stated

*Response:* *We agree with reviewer's comment and have removed this sentence "We refer to the combination of sensors and monitors as hybrid instruments".*

9. Line 67: "noble" should be "notable"

*Response:* *We agree with the reviewer's comment and have replaced the term "noble" with "notable" in line 65 of page 3.*

10. Line 70: "Therefore, following" should be "Therefore, the following"

*Response:* *We agree with the reviewer's comment and have replaced the term "Therefore, following" with "Therefore, the following" in line 68 of page 3.*

11. Line 77: "Next section" should be "The next section"

*Response:* *We agree with the reviewer's comment and have replaced the term "Next section" with "The next section" in line 75 of page 3.*

12. Line 80: "of hybrid" should be "of a hybrid"

***Response:*** *We agree with the reviewer's comment and have replaced the term "of hybrid" with "of a hybrid" in line 78 of page 3.*

13. Line 84: Consider restating the objective to better explain "people satisfaction", e.g., "Our approach focuses on placing sensors in order to maximize a utility function quantifying popular satisfaction with the sensor placements".

***Response:*** *We agree with the reviewer's comment and have replaced the term "The approach focuses on the utility gain of placement of sensors as per people satisfaction." with "Our approach focuses on placing sensors and monitors in order to maximize a utility function quantifying popular satisfaction with the instrument placements." in lines 82-83 of page 3.*

14. Line 87: "g(d) be" should be "g(d) must be"

***Response:*** *We agree with the reviewer's comment and have replaced the term "g(d) be" with "g(d) must be" in line 88 of page 3.*

15. Equation 1: describe how the parameter theta is set

***Response:*** *We thank the reviewer for his/her valuable feedback. By introducing theta as a decaying parameter of distance, we effectively control the rate at which g(d) decreases as the distance increases. Depending on the largest distances that are considered in a grid network and the precision that is being considered, $\theta$ should be appropriately decided. For instance, if the computation precision being used is say about $10^{-5}$ and the largest distance is say 10 units then $\theta = 1$ might reasonable since $e^{-\frac{10}{1}} = 4.5 * 10^{-5}$. This has now been added as a footnote on page 4.*

16. Line 101: While emissions have an influence on local PM2.5 concentrations, secondary aerosol production and pollution transport also play a role.

***Response:*** *We agree with reviewer's comment. In our formulation, we focus on PM2.5 emissions as a starting point for network design. The goal is to strategically deploy sensors and monitors in an area to identify and analyze sources of pollution. We have now the changed the sentence "PM2.5 emission indicates the level of fine particulate matter in the air in that grid" with "PM2.5 emissions are an indicator of the level of fine particulate matter in the air within that grid (secondary aerosol production and pollution transport also play a role in the concentrations but they are not considered here due to lack of data)." on line 111-113 on page 4.*

17. Lines 106-107: Move this sentence right after the first one in this paragraph.

*Response: We thank the reviewer for his/her valuable insight and hence shifted the sentence "The notations are summarized in Table 1 of appendix." after the first sentence in this paragraph in lines 118-119 on page 4.*

18. Line 110: "where monitor" should be "where a monitor"

*Response: We agree with the reviewer's comment and have replaced the term "where monitor" with "where a monitor" in line 124 of page 5.*

19. Equation 4: It is not clear why sensors deployments should be required, but monitor deployments should not be.

*Response: We thank the reviewer for inquiring regarding the Equation (4) in Problem Statement. The locations for sensors could be hospitals, nursery, malls, market or crowded area etc. where authorities want to know the air quality of that place. However, monitor deployment has not been considered that flexibly because monitors cannot be place anywhere, they need to be set a specific place where electricity is available, they are big and heavy as compared to sensors, skilled engineers would be required for their maintenance. Also, monitors are much costlier and so we cannot put them everywhere. We have added this in lines 135-137 of page 5.*

20. Equation 5: Similarly, it is not. clear why monitor deployments are restricted, but sensor deployments are not.

*Response: We thank the reviewer for this comment. The potential locations for placing instruments can be places with sparse population, water bodies, etc. However, it may not be cost-effective or practical to deploy expensive monitors in such areas and thus monitor deployments are restricted, but sensor deployments are not. The above stated sentence have been added to the manuscript in lines 138-140 of page 5.*

21. Line 115: Please define d(a,b).

*Response: Thank you. We define d(a, b) as the distance between grid a and grid b. We have now provided a clearer definition of d(a, b) in lines 131-132 of page 5.*

22. Line 129: "or" should be "of"

*Response: We agree with the reviewer's comment and have replaced the term "or" with "of" in line 148 of page 6.*

23. Line 156: "carried" should be "carried out"

*Response: We agree with the reviewer's comment and have replaced the term "carried" with "carried out" in line 174 of page 7.*

24. Line 164: describe how the parameter alpha was chosen

*Response: We thank the reviewer for this comment. The choice of the value alpha, set to 10^(-5), was made based on extensive experimentation and consideration of the algorithm's behavior in our optimization framework. Alpha serves as a stopping criterion in our genetic algorithm, influencing its convergence behavior. This specific value was determined through a systematic tuning process. We initially experimented with a range of alpha values, including both higher and lower values, to observe their impact on convergence and the quality of solutions generated. After careful assessment and analysis of multiple runs, we found that alpha = 10^(-5) consistently produced satisfactory convergence behavior while maintaining computational efficiency. The above stated explanation has been added in lines 268-270 of page 11.*

25. Line 171: I believe that a maximization expression is missing in the equation here.

*Response: Thank you. We have now written $s^* = argmax_s \sum_{a=1}^{n} m_a \left( g\big(d'(a, K \cup s)\big) - g\big(d'(a, K)\big) \right)$ in line 189 of page 7.*

26. Line 175: "reduce" should be "subtract"

*Response: We agree with the reviewer's comment and have replaced the term "reduce" with "subtract" in line 193 of page 7.*

27. Line 179: Stopping criteria are not described for the greedy algorithm.

*Response: Thank you. The greedy algorithm stops whenever P' ≈ 0 or there is not enough amount of budget left for the placement of hybrid equipment that is considered as stopping criteria. A sentence has been added in the manuscript in lines 197-198 of page 7.*

28. Line 190: Please provide citations or links to the World Bank and TERI datasets used here.

*Response: Thank you. Our population data was obtained from an open source site called WorldPop and we have now provided the link for that. We have now provided this as a footnote in page 9. However, the emission data from TERI was obtained after our request and is not available as an open source. However, we have now also provided both the population densities and emissions used for different grids in Surat (please see Figures 2, 3 and 4 on page 10) and the population densities used in Mumbai (please see the footnote on page 18).*

29. Line 220: "of the Mumbai" should be "of Mumbai"

*Response: We agree with the reviewer's comment and have replaced the term "of the Mumbai" with "of Mumbai" in line 357 of page 16.*

30. Line 224: "maintained consistently as above in" should be "the same as in the example for"

*Response: We agree with the reviewer's comment and have replaced the term "maintained consistently as above in" with "the same as in the example for" in lines 361-362 of page 16.*

31. Line 225: "we have larger number" should be "we have a larger number"

*Response: We agree with the reviewer's comment and have replaced the term "we have larger number" with "we have a larger number" in line 363 of page 16.*

32. Line 244: "solution is" should be "solution that is"

*Response: We agree with the reviewer's comment and have made the change in line 383 of page 17.*

33. Figure 5: It is unclear how the size of the grids is being varied; Is this the same example for Mumbai? Are the sizes of grids being reduced, or is the area of coverage being increased?

*Response: We thank the reviewer for the comment. In Figure 12 we illustrate the impact of changing the area of coverage rather than altering the sizes of the grids. Specifically, the size of the grids has been held constant at 1 km x 1 km throughout the example for Mumbai. The variation in Figure 12 pertains to the expansion of the area covered by our monitoring network, which involves increasing the number of grids. This expansion allows us to investigate how our optimization framework performs as the coverage area grows.*

34. Table 1: Describing g(d) as a function of d is not very informative; consider expanding the description and referring back to Equation 1 for the definition.

***Response:*** *We thank the reviewer for pointing out this. Now we have defined the g(d) function in Table 2 as "an individual's satisfaction as a function of his or her distance d to the closest sensor or monitor".*

35. References: It appears that a citation is missing for Lerner et al. 2019

***Response:*** *We thank the reviewer for this comment. In response to your feedback, we have added the reference to Lerner et al. (2019) in the reference section (see page 22).*

---

## Author Comment (AC2)

**Response to Reviewer 2 comments on "**Hybrid Instruments Network Optimization for Air Quality Monitoring**" (Manuscript** AMT-2023-173**) submitted to *Atmospheric Measurement Techniques***

**Reviewer 2**

(1) The rationale for the methodology is insufficiently explained.

-The authors do a nice job of explaining their methods, however they don't give a lot of explanation for why they made specific choices. For example, the optimization relies on the satisfaction function "g". The authors assume that "g" increases as a person is placed closer to the nearest monitor, however that assumption is not described or justified in great detail. For example, is it universally true that g increases as distance decreases? For a hybrid network, will people treat both monitors and sensors as equally valuable, and therefore g(0) = 1 for both sensors and monitors?

*Response: We appreciate the referee's feedback. We have taken the following steps to address this concern:*

1. *Clarification of the Satisfaction Function "g": We agree that the assumption regarding the satisfaction function "g" should be better justified. In our revised manuscript, we provide a detailed explanation for the choice of "g" as a function that increases as a person is placed closer to the nearest monitor. Note that people will have higher confidence on the readings by sensors or monitors that are closer to them rather than readings from instruments that are farther from them. The rationale for this assumption is supported by papers such as (Sun et al., 2019)[1]. Please check lines 84-87 of page 3.*

2. *Hybrid Network Considerations: We recognize that the referee raises a valid point about treating monitors and sensors equally in a hybrid network. We accept that satisfaction should not be the same for sensors and monitors if the accuracy of these instruments is known to the users and our optimization formulation can be modified to consider that. However, in many practical air quality monitoring scenarios, users may*
* * *
[1] Sun, C., Li, V.O., Lam, J.C. and Leslie, I., 2019. Optimal citizen-centric sensor placement for air quality monitoring: a case study of city of Cambridge, the United Kingdom. IEEE Access, 7, pp.47390-47400.

*not be either interested or be able distinguish between data collected from monitors and sensors (if the information related to the type of instrument is not openly available). From the user's perspective, the primary concern may be just to obtain reasonable air quality information, rather than worry about the specific source of the data. That is why, in our case we have only considered the difference of cost between sensors and monitors. In summary, the choice to consider monitors and sensors to be equivalent apart from the cost (and thus g(0)=1 for both the instruments) is rooted in practical considerations, as the users may prioritize air quality information over the source of the data. We have now added this in lines 98-102 on page 4.*

**Reviewer 2**

If we accept that g decreases with increasing distance (which seems reasonable), the selection of the exponential decay for g(d) seems arbitrary. Other functions would fit the rules set by the authors, and they don't explore those as sensitivity cases or explain why an exponential function is the most likely one.

***Response:*** *We thank the referee for his comment. The exponential decay function is often chosen in similar studies and practical applications because of its simplicity and effectiveness in modeling the attenuation of signal or influence with increasing distance in studies such as Sun et al. (2019). It aligns with the intuitive idea that the influence of air quality monitoring decreases as one moves farther away from the monitor. We have now added this in lines 94-96 on page 4.*

*We appreciate the referee's suggestion to explore alternative functions as sensitivity cases. As previously mentioned, the g(d) function should be a decreasing function. Therefore, we explore an alternative function $g(d) = \frac{1}{d+1}$ apart from the exponential function. We have now obtained the results by greedy algorithm and genetic algorithm for Surat city grid using $g(d) = \frac{1}{d+1}$, while by keeping all the other parameters the same. The following figure (put as Fig. 8 in the revised manuscript) shows the results with different functional forms of $g(d)$.*

[Figure]

*It can be seen in the above figure that the values obtained by the genetic algorithm and the greedy algorithm for $g(d) = e^{-d}$ are very close and thus the solid blue and red curves almost overlap. The same holds for $g(d) = \frac{1}{d+1}$ and therefore the dashed black and purple lines almost overlap. Note that the values that are obtained by the two algorithms for $g(d) = \frac{1}{d+1}$ are greater than that obtained for $g(d) = e^{-d}$. That is because $\frac{1}{d+1} > e^{-d}$ for all positive values of $d$. However, notice that the pattern of the values that are obtained for the two functional forms is the same, i.e., the values decreases as the total available budget increases. Also, notice that the values obtained by the two functional forms converge at the budget value of \$313,000. Since it is not possible to have percentages values greater than 100, the values for both the functional forms will remain the same for budget values greater than \$313,000. We believe that similar patterns will be observed by other functional forms of $g(d)$ as long as they satisfy the conditions that are necessary for satisfaction functions (i.e., g(d) must be a strictly decreasing function and g(0)= 1). This has now been added in Section 3.1.1.1 on pages 13-14.*

**Reviewer2**

Overall, the methods are presented well but the rationale is not given. For example, the authors state that they average the fractional population and fractional emissions in each grid. However the basis for this averaging is not given, and it's not clear how this averaging helps the model output (and indeed this averaging is not done for the Mumbai case).

*Response: We thank the referee for this feedback. The justification for averaging PM2.5 Emissions and Population Density is as follows:*

1. *Balancing Priorities: PM2.5 emissions and population density are two essential factors for air quality sensor placements. By averaging these variables, we strike a balance between the need to monitor areas with high pollution levels (captured by PM2.5 emissions) and areas with high population density (captured by the population density). The above justification has been added in line 262-264 of page 11.*

2. *Dimensionality Reduction: Averaging reduces the dimensionality of the input data, making it more manageable for optimization algorithms. That is, we are able to focus on an optimization problem that has a single objective function due to the averaging process. Had we not done the averaging, and targeted to individually minimize some metrics related to emission and population then it would result into a multi-objective optimization problem which is much more difficult to solve and analyze (Deb, 2001)[2]. This has been added in lines 113-116 of page 4.*

*Also, averaging is not done for Mumbai due to the absence of $PM_{2.5}$ data for Mumbai. However, this is not a limitation of the methodology that we propose and if $PM_{2.5}$ emissions data becomes available for Mumbai then we will use the same method as that we used for Surat.*

*We have also done the sensitivity analysis by changing the weightage between the percentages of population density and $PM_{2.5}$ emissions for Surat city (5 km x 5 km area). Greedy algorithm and genetic algorithm are used for this sensitivity analysis by keeping all the parameters same.*

***Different cases for the weights***

| Case | Weightage for $p_a$ | Weightage for $e_a$ |
|------|---------------------|---------------------|
| 1 | *0.25* | *0.75* |
* * *
[2] Deb, K., 2011. Multi-objective optimisation using evolutionary algorithms: an introduction. In Multi-objective evolutionary optimisation for product design and manufacturing (pp. 3-34). London: Springer London.

| 2 | 0.5 | 0.5 |
| 3 | 0.75 | 0.25 |

[Figure]

*The above figure shows the values that are obtained for different cases, budget values and algorithms. As before, the values obtained by GA and GrA are very close for given weights and budget. Among these cases, the values corresponding to Case 3 (where $p_a = 0.75$ and $e_a = 0.25$) are the highest and that corresponding to Case 1 (where $p_a = 0.25$ and $e_a = 0.75$) are the lowest. Thus, as the relative weightage for population density increases in the objective function, the values obtained increases. However, it can be seen that the difference between the values for Cases 1 and 3 is not that large, signifying that the objective function values may not be that sensitive to the relative weightage between population density and emissions. This discussion has now been added in Section 3.1.1.2 of pages 14-15.*

**Reviewer 2**

Section 2.1 could benefit from some sort of schematic would make this all much easier to follow. A visual representation of what the optimization is trying to do would help readers follow along.

*Response: We agree that a visual representation can significantly enhance the understanding of the optimization process. To address this suggestion, we will take the following steps in the revised manuscript:*

*We now provide an example of a 3x3 network (i.e., a network having 3x3 = 9 grids) to illustrate the greedy algorithm. The population density data and PM2.5 emissions data for a 3x3 network are provided below on the left and right, respectively.*

| | | |
|---|---|---|
| 65646 | 29660 | 15504 |
| 9487 | 2984 | 2260 |
| 2042 | 2393 | 1711 |

| | | |
|---|---|---|
| 0.143405 | 0.120589 | 0.097773 |
| 0.114025 | 0.142434 | 0.170843 |
| 0.084646 | 0.16428 | 0.243914 |

*Then we calculate the percentage of population density ($p_a$) and PM$_{2.5}$ emissions ($e_a$) for each grid and then calculate $m_a$ which is an average of $p_a$ and $e_a$. The following tables show the values of $p_a$ (left) and $e_a$ (right).*

| | | |
|---|---|---|
| 49.85 | 22.5231 | 11.7734 |
| 7.2042 | 2.266 | 1.7162 |
| 1.5506 | 1.8172 | 1.2993 |

| | | |
|---|---|---|
| 11.1868 | 9.407 | 7.6271 |
| 8.895 | 11.111 | 13.3273 |
| 6.6031 | 12.8153 | 19.0274 |

*The following values are the $m_a$ values for each grid of the 3x3 network that we consider.*

| | | |
|---|---|---|
| 30.5184 | 15.965 | 9.7002 |
| 8.0496 | 6.6885 | 7.5217 |
| 4.0769 | 7.3162 | 10.1633 |

*Suppose the set B in which at least one sensor is to be placed from Equation (4) is consists of grids 7 and 9 and set C in which no monitor can be placed from Equation (5) is given by set 7. Suppose $h = 2$, which represents the minimum number of monitors required. Let the cost of sensor (c) and monitor (c') be 200 and 8000 units respectively. The total available budget be 16500 units.*

[Figure]

***Fig. 1 Example to show the working of greedy algorithm***

*Figure 1p) shows the initial empty grids which are grey in color. Figure 1q) shows the grid area in which two grids (i.e., grids 7 and 9) are shown in light green color grids which tells us about grids in set B, where at least one sensor must be placed. Given that the value of $m_a$ for grid 9 is greater than that of grid 7, a sensor is initially placed in grid 9 to satisfy Equation (4). The placement of a sensor at grid 9 reduces the available budget to 16300 units.*

*Figure 1r) shows the placement of sensor at grid 9 and a grid (corresponding to set C) which is shown by orange colored square grid (i.e., grid 7). The monitors are positioned at grid 1 and 2 based on the values obtained from the largest information gain $s^*$ and in adherence to the Equation (5) which has a requirement that no monitor be placed on any grid belonging to set C. This further reduces the budget from 16300 units to 300 units by subtracting 16000 units (i.e., $c'h$)*

*We continue to place sensors until the budget constraint is violated. We will place next sensor at the grid with largest information gain $s^*$ and that grid is grid 3. This further reduces the budget from 300 units to 100 units. The algorithm stops here as there is no sufficient budget to proceed. Figure 1s) shows the final solution using greedy algorithm where grey colored square grids show the empty grids, purple colored square grids shows the placement location of sensors and light yellow colored square grids shows the placement location of monitors.*

*This example has now been added in pages 7-9.*

**Reviewer2**

Lastly, the context for the two case studies is not presented very well. For example, Fig 1 shows a map of the predicted sensor locations for Surat. However, we are not provided with any

additional information, such as the population density or the emissions locations, that would help us to understand the results.

*Response: We appreciate the referee's feedback. We acknowledge the importance of providing a comprehensive understanding of the study context, and thus we are providing the data for population density and PM2.5 emissions that we used.*

*The total number of grids in Surat are 25 which are numbered from 1 to 25 from left to right in increasing order and from top to bottom in increasing order. This is shown in the figure below and has been added as Figure 2 of the revised manuscript.*

| 1 | 2 | 3 | 4 | 5 |
|---|---|---|---|---|
| 6 | 7 | 8 | 9 | 10 |
| 11 | 12 | 13 | 14 | 15 |
| 16 | 17 | 18 | 19 | 20 |
| 21 | 22 | 23 | 24 | 25 |

*The population density data for Surat network is as follows (in population per sq. km) and has now been added as Figure 3 of the revised manuscript.*

| 44252 | 74524 | 85060 | 66989 | 94922 |
|---|---|---|---|---|
| 23631 | 50185 | 74016 | 80964 | 86887 |
| 40666 | 69841 | 65646 | 29660 | 15504 |
| 29549 | 21068 | 9487 | 2984 | 2260 |
| 4267 | 2293 | 2042 | 2393 | 1711 |

*The PM2.5 emissions data for Surat network is as follows (in kT/yr) and has now been added as Figure 4 of the revised manuscript.*

| 0.29385 | 0.497288 | 0.700726 | 0.665802 | 0.630877 |
|---|---|---|---|---|
| 0.199782 | 0.310924 | 0.422065 | 0.393195 | 0.364325 |
| 0.105715 | 0.12456 | 0.143405 | 0.120589 | 0.097773 |
| 0.056277 | 0.085151 | 0.114025 | 0.142434 | 0.170843 |
| 0.006839 | 0.045742 | 0.084646 | 0.16428 | 0.243914 |

The link for population data for Mumbai network is now available in the footnote of page 18.

**Reviewer 2**

(2) The results are not described in sufficient detail.

Figure 1 looks like each algorithm suggests even spacing. Isn't this basically the null hypothesis? None of the discussion of the results explains how the GA and GrA output differ, or why they reach slightly different network designs. Why does one algorithm place a reference monitor in the extreme southwest of the domain but the other places them both closer to the center?

*Response: We appreciate the referee's feedback. The even spacing of monitors and sensors in Figure 5 is not the null hypothesis instead we have assumed that sensors and monitors are placed evenly and in the center of grids for the two case studies that we have considered. However, the optimization formulation does not require the instruments to be evenly spaced. The satisfaction function only depends on the distance $d$ and thus our formulation will work even if we have unevenly spaced grids. The reason behind the use of square grids in the case studies is that the data that we use on population density and PM2.5 emissions was collected at a 1 km x 1 km scale. Also, logically it makes sense to place an instrument at the center of a gird. Thus, because of square grids and centers being used for grids, we get even spacing between consecutive potential locations for instruments.*

*The greedy algorithm is based on the principle of making the locally optimal choice at each step with the hope of finding a global optimum. While Greedy Algorithm doesn't guarantee an optimal solution for all problems, it is a very efficient algorithm as it deterministically determines a solution. In contrast, the Genetic Algorithm is a probabilistic algorithm that applies the stochastic operators such as selection, crossover and mutation to improve a set of solutions across different iterations In Figure 5, the Greedy Algorithm (GrA) places monitors close to each other. That is because after placing one sensor at a grid in set B, the algorithm positions monitors at grids with the highest $s^*$ values. This leads to monitors being placed close together, as seen at grids 8 and 12. In contrast, the solutions of the Genetic Algorithm are generated through a probabilistic process and thus may exhibit a different spatial distribution than that obtained by the Greedy Algorithm. Note that the objective function value corresponding to both the algorithms for this case is equal to 100 (see Figure 6) but the spatial*

*distribution of the instruments is not the same. That is because this is a discrete optimization problem and it can also be possible that two solutions with very different looking spatial distribution can have the same objective function value. We have added this in lines 255-262 of page 11.*

**Reviewer 2**

Similarly, the GA and GrA put monitors/sensors in the same locations in Mumbai, with the only difference being the locations of the reference monitors. Again, why is this the only difference?

*Response: We thank the referee for this comment. While it may appear that the primary distinction is in the locations of the reference monitors, there are also other differences between the solutions obtained by the two algorithms, as explained next. In the GA results for Mumbai, two sensors are positioned in the northeast area, while no sensors or monitors are placed in that area in GrA. In GrA, monitors/sensors are predominantly concentrated on the left side of the Mumbai area, whereas in GA, the sensors/monitors exhibit a more diverse and scattered distribution. Note that out of 100 grids, sensors and monitors can be placed in only 15 grids by maximizing the objective function. The leftmost and southern areas have the highest population density, which explains the concentration of sensors and monitors in those regions. The above stated paragraph has been added in the manuscript in lines 378-385 of pages 16-17. Note that as mentioned in the previous response, these two algorithms have different characteristics and therefore the solutions obtained by them need not be the same.*

**Reviewer 2**

Each case study has a specified maximum budget. However, the rationales for these budgets is never stated; they are just presented as part of the results.

*Response: We thank the referee for this comment. We have given the rationale on the specific budget values in lines 274-275 of page 11 and have added a sentence in the manuscript which has been show in red font here. It is stated that "The minimum budget that is considered is $253,000, which is equal to the cost of three sensors plus h monitors* *(any value of budget lower than this will not yield a feasible solution of the problem as other constraints will not get satisfied)**. The maximum budget in Figure 2 is $313,000, which allows for the placement of 2 monitors and 23 sensors, covering the entire portion area (as there are a total of 25 grids) under minimum possible budget as at least 2 monitors need to be placed by Equation (6)."If*

*we keep on increasing the budget, then it might be possible that monitor is increased from 2 to 3 and so on (but that would not yield any increase in the objective function value as satisfaction function is assumed to be identical for sensors and monitors).* The previous sentence has been added in lines 278-280 of page 11.

**Reviewer 2**

Figure 2, 3, and 5: I think the "value" on the left axis is the value g used to determine the value of each sensor, but that is not clear. Furthermore, the function g has values 0-1 inclusive, so why do these plots have a range from ~60-100?

***Response:*** *We thank the referee for asking this question. We would like to tell that the obtained value on the left axis in Figure 6, 8, 9, 10 and 12 is the objective function value which is the summation of the products of $m_a$ and $g(d)$ over all grid points. The definition of "Obtained Value" has been defined in line 273 of page 11. Therefore, these plots have a range from ~60-100.*

**Reviewer 2**

Line 195-200 - references on the costs for sensors and monitors would be nice. What types of sensors and monitors are being considered?

***Response:*** *We thank the referee for this comment. We obtained the cost estimate for a monitor through the cost of imported continuous ambient air quality monitoring stations (CAAQMS) whose price is available at the following link: [https://timesofindia.indiatimes.com/india/centre-asks-states-not-to-procure-imported-air-quality-monitors-indigenous-systems-to-be-deployed/articleshow/95901936.cms](https://timesofindia.indiatimes.com/india/centre-asks-states-not-to-procure-imported-air-quality-monitors-indigenous-systems-to-be-deployed/articleshow/95901936.cms) . Similarly, the cost of a sensor (here Aeroqual S500) is estimated from the following link: [https://www.cleanair.com/product/aeroqual-s500-starter-kit/](https://www.cleanair.com/product/aeroqual-s500-starter-kit/). In the revised manuscript, we have included this as a footnote on page 11.*

**Reviewer 2**

Line 225 - I don't think that the variable theta is defined above

***Response:*** *We thank the referee for asking this question. We would like to tell that variable theta is defined in Section 2.1 Problem statement in lines 94-97 of page 4.*

**Reviewer 2**

(3) There is a lack of sensitivity and generalizability

-The authors seem to have solved a very narrowly defined problem for two specific cases. While that is fine, they don't discuss how the results of this paper can be used more broadly to inform sensor network design. For example, what about the optimization for Mumbai could be useful to users who are building a network in another location (and perhaps don't have the ability to run this algorithm)?

*Response: We appreciate the referee's feedback. In response to this comment, we would like to address the following points: Our paper primarily focuses on the optimization of air quality monitoring networks in Surat and Mumbai cities, which are used as illustrative cases. While the study's results are specific to these locations, the underlying methodology and principles learned from these cases can be broadly applied to other areas facing similar air quality monitoring challenges. The methodology presented in our paper serves as a template for optimizing sensor networks in any location, provided that relevant data on population, emissions, and potential grid locations are available. We emphasize in the revised manuscript that the same approach can be adopted by researchers, policymakers, and practitioners in other regions to design effective sensor networks based on local data. The above discussion has been added in the manuscript in lines 421-424 of page 19.*

*We also mention the potential for creating user-friendly software tools or decision support systems based on the methodology presented in our paper. Such tools would enable users with limited algorithmic expertise to apply similar optimization techniques to their specific locations, addressing the concern of not having the ability to run the algorithm. In these software tools, the users will only have to provide input values for the problem like the network they want to solve, costs of instruments, budget, the algorithm they want to use, etc., and the toolbox will provide the results. The above paragraph has been added in lines 447-452 of page 20.*

**Reviewer: 13 General Comments:**

The authors do very little to test the sensitivity of their optimized network. Figure 5 shows how the results change with the number of grids in Mumbai, but they do not seem to probe the results further than that. For example, how does the functional form of "g" impact the results? How does the averaging of population and emissions (or lack thereof) impact the results? How much "better" does an optimal network perform compared to one where all of the sensors cannot be placed exactly at the optimal locations? I think that without probing the output further and providing more information on the broader impact of this work, the paper will have very little impact.

*Response: We appreciate the referee's feedback. We provide a more comprehensive analysis in our revised manuscript to address these concerns:*

*Sensitivity Analysis of the Functional Form of "g": We conduct a sensitivity analysis to explore how a functional form of $g(d)$ other than the exponential form impacts the network optimization results. This involves examining the effects of another distance-decay function $g(d) = \frac{1}{d+1}$ on the performance of the sensor network. We quantify the differences and discuss their implications to provide a clearer understanding of the sensitivity of our model to this function. The sensitivity analysis has been in Section 3.1.1.1.*

*Impact of Averaging Population and Emissions: We delve deeper into the impact of averaging population and emissions within grid cells. By comparing the results with scenarios that do not consider equal weights for emissions and population, we now assess how this specific aspect of our methodology affects the optimized network. This sensitivity analysis has been added in Section 3.1.1.2. We have also now better justified the need for computing a weighted average of the percentages of population density and emissions in lines 108-116 of page 4.*

*Performance Comparison with Non-Optimal Sensor Placements: We now include a comparative analysis between the performance of different network configurations that have different variations with respect to the optimal solution.*

*Consider an example of 3 x 3 grid. Let the cost of sensor ($c$) and monitor ($c'$) be $3000 and $122000 respectively. Suppose the budget value is equal to $253000. The numbering of the*

grids follows the convention that numbers first increase as we go from left to right in the increasing order and numbers increase as we go from top to bottom. Let the set $B$ in which at least one sensor is to be placed from Equation (4) consist of grids 7 and 9 and set $C$ in which no monitor can be placed from Equation (5) be set 7. We consider four different feasible solutions as follows:

Case 1: Solution obtained from greedy algorithm.

[Figure]

The above figure shows the solution that is obtained in Case 1. The purple points show the placement location of sensors and orange points show the placement location of monitors. It can be seen that monitors are placed at grids 1 and 2 and sensors are placed at grids 3, 4 and 9.

Case 2: Sensor placed at grid 3 in Case 1 is moved to grid 7 (all the other instrument locations remain the same as in Case 1).

Case 3: Monitor placed at grid 1 in Case 1 is moved to grid 5 (all the other instrument locations remain the same as in Case 1).

Case 4: When sensor placed at grid 3 in Case 1 is moved to grid 7 and monitor placed at grid 1 in Case 1 is moved to grid 5 (all the other instrument locations remain the same as in Case 1).

The following table shows the values that are obtained for different cases.

| Case | Obtained Value |
|--------|----------------|
| Case 1 | 83.8154 |
| Case 2 | 80.2608 |

| Case 3 | 68.7521 |
|--------|---------|
| Case 4 | 65.1975 |

*It can be seen that Case 1 has the largest value and the values decrease as we go from Case 1 to Case 4. Thus, Case 1 is the closest to the optimal solution and Case 4 is the farthest. Note that Case 4 has both the modifications that are made in Cases 2 and 3 with respect to Case 1. Since there was decrease in the value as we go from Case 1 to Case 2 and from Case 1 to Case 3, the largest decrease is seen as we go from Case 1 to Case 4. The above discussion has been included in lines 292-315 of pages 12-13.*

---

## Author Response (AR2)

**Response to Reviewer comments on "Hybrid Instruments Network Optimization for Air Quality Monitoring" (Manuscript AMT-2023-173) submitted to *Atmospheric Measurement Techniques***

**Reviewer**

In the new sensitivity studies presented in 3.1.1.1 and 3.1.1.2, while you compare the objective values and trends using the different definitions in the objective, the key question to investigate is how the placement of sensors varies (or does not vary) between these different approaches. To do this, you should define similarity metric (e.g., fraction of the grid cells with instruments which are the same in both approaches). You may need to run multiple trials with the genetic algorithm to get robust results, as the selected locations will be somewhat randomized in that approach.

*Response: We appreciate the referee's feedback. We have now defined a similarity index that quantifies the difference in the placement of hybrid instruments as obtained by different algorithms. Suppose the number of grids where the placement of hybrid instruments is identical is given by k (a grid is said to have identical placement by the two algorithms if the grid contains a sensor as determined by both the algorithms or a monitor as determined by both the algorithms). Also, let the maximum number of hybrid instruments that can be placed in the given constraints be equal to p. Then, similarity index is given by k/p. Since the solution obtained by the genetic algorithm is probabilistic, we tested five runs of genetic algorithm (for a given budget value) and compared the solution obtained by each run to the solution obtained by the greedy algorithm to determine the similarity indices and finally obtained the average similarity index by taking the mean of five similarity indices. The following figure shows the average similarity index for different budget values and for different $g(d)$ functions (while keeping equal weights for the percentages of population density and emissions). Note that similarity index is upper bounded by one. Also, we see that as budget values increase the average similarity index for both $g(d)$ functions increase. That is because as the budget increases the number of grids at which instruments can be placed increases and both the algorithms usually place sensors at most grids except at a few grids where monitors are placed to meet the requirement of minimum monitors. Note that the average similarity index is around 0.5 for low budget values due to the existence of solutions with that have varying placement but have close objective function values (but as the budget increases the variation in the*

*placement reduces as explained before). Also, the average similarity indices obtained by the two $g(d)$ functions are close for most of the budget values. The figure after that shows the average similarity index for different budget values and different cases corresponding to the weights of percentages of population density and PM2.5 emissions (while keeping $g(d)$ as exponential function). Table 2 in the paper shows the weights corresponding to different cases. It can be seen that the average similarity index increases with budget values for the same reason mentioned before. Also, the values of similarity indices are close for most of the budget values. These two figures are in the revised manuscript as Figures 5 and 7 and this discussion is on pages 12 and 14.*

[Figure]

[Figure]

**Reviewer**

Lines 100-104: While I agree that the public might not distinguish between monitors and sensors in terms of data quality, the designer of the network should. This is a reason why additional objective might be included in the optimization related to the distribution of the monitors, e.g. so that they provide a robust baseline value for the rest of the network. Further, if there is no distinction between these from the point of view of the optimization algorithm, it will always choose the cheaper sensors, once the minimum number of monitors has been placed.

**Response:** *Thank you. We agree that the designer of the network may want to distinguish between sensors and monitors although the public may not distinguish. Therefore, we now provide an alternate optimization formulation whose objective is to maximize the weighted sum of satisfaction functions from monitors and sensors. Let $w_s$ be the weight corresponding to the satisfaction from sensors and $w_m$ be the weight corresponding to the satisfaction from monitors. Let $d(a)$ be the minimum distance between grid $a$ and any grid containing sensors and $d'(a)$ be the minimum distance between grid $a$ and any grid containing monitors. The remaining parameters and variables mean the same as before. Then, the formulation is as follows:*

$$\max w_s \sum_{a=1}^{n} m_a \cdot g\big(d(a)\big) + w_m \sum_{a=1}^{n} m_a \cdot g\big(d'(a)\big) \qquad \text{(B1)}$$

$$\text{s.t. } \sum_{a=1}^{n} (cx_a + c'y_a) \leq P \qquad \text{(B2)}$$

$$\sum_{a \in B} x_a \geq 1 \qquad \text{(B3)}$$

$$\sum_{a \in C} y_a = 0 \qquad \text{(B4)}$$

$$\sum_{a=1}^{n} y_a \geq h \qquad \text{(B5)}$$

where $d(a) = \min_{b \in V} \{x_b \cdot d(a,b) + \bar{d}(a) \cdot (1 - x_b)\}$,

$\bar{d}(a) = \max_{b \in V} d(a,b)$ and $d'(a) = \min_{b \in V} \{y_b \cdot d(a,b) + \bar{d}(a) \cdot (1 - y_b)\}$.

*Thus, the relative values of the weights $w_s$ and $w_m$ decide the relative importance being given to monitors and sensors. Typically, $w_m$ should be chosen larger than $w_s$ as monitors are more accurate than sensors. One could solve the above formulation with minimal changes to the proposed genetic and greedy algorithms. This formulation is now provided in Appendix B of the revised manuscript.*

*Also, we agree that in the original formulation since there is no distinction between sensors and monitors, the optimization algorithms will usually select sensors after fulfilling the requirements of minimum number of monitors since sensors are cheaper. That is why in the greedy algorithm that we proposed, we select sensors (until the budget allows) once the constraints (4), (5) and (6) are met. Please check lines 199-202 on pages 7 and 8.*

**Reviewer**

Lines 205-233: Suggest that this example be relocated to the appendix.

**Response:** *We thank the reviewer's feedback and have now shifted this example to Appendix C of the revised manuscript.*

**Reviewer**

Figures 3 and 4: Suggest these be presented as shaded grids where the shading intensity is proportional to the value. Ideally, these could also be overlaid on the map of Surat City.

**Response:** *We thank the reviewer's feedback. We have now presented these grids with shades that reflect the respective intensities. Please check Figure 1 in the revised manuscript.*

[Figure]

**Reviewer**

Lines 264-267: The differences in placement are immaterial; as the sensors and monitors provide the same value, any placement covering all 25 grid cells which matches the constraints will have the same value. In the case of the greedy algorithm, at least, since the monitors are placed first, they represent the highest-valued pair of locations. The Genetic algorithm will have no such prioritization. It might be more illustrative to present a result where the budget only allows about half of the cells to contain instruments; comparing these might reveal more about the different placement approaches of the algorithms. Still, any difference in the

placement of monitors v. sensors in the GA approach will be mostly random. This should be noted in the discussion of results.

**Response:** *We thank the reviewer's feedback. We have now replaced Figure 2 to show the results corresponding to the budget value equal to $295000 (instead for the budget of $313000 that was shown earlier). Now all the 25 girds are not covered with instruments (which is the case when the budget is $313000). Also, the solutions obtained by both the genetic and greedy algorithms have two monitors and seventeen sensors although the spatial placement of these instruments is not identical (it is possible that two solutions with very different looking spatial distribution can have the same objective function value). Note that there is no scope to further add any instrument in the solution of any of the algorithms as 2\*122000 + 17\*3000 = 295000. However, if the budget is sufficiently large and the optimal solution involves covering all the grids then genetic algorithm can provide solutions where at some places interchanging sensors with monitors will not change the value of the solution (because the objective function does not differentiate between monitors and sensors). We have now mentioned this point in lines 229-234 of page 9 and in lines 250-253 of page 10.*

[Figure]

[Figure]

**Reviewer**

Lines 300-324: The motivation for presenting this is not clear to me. You seem to be illustrating that the sub-optimal solutions are not optimal, which is intuitive. This may also be better suited for the appendix.

**Response:** *Thank you. We have now shifted this example to Appendix D of the revised manuscript.*

**Reviewer**

Line 328: I suggest not putting the equation in the section heading.

**Response:** *Thank you. We have now replaced '3.1.1.1 Results for g(d)=1/(d+1)' to '3.1.1.1 Sensitivity analysis with another g(d) function' at line 269 on page 11 of the revised manuscript.*